# EQUIVARIANT SPLITTING: SELF-SUPERVISED LEARNING FROM INCOMPLETE DATA

Victor Sechaud[*1], Jérémy Scanvic[*1, 2], Quentin Barthélemy[2], Patrice Abry[1], and Julián Tachella[1]

[1]LPENSL, CNRS, ENS de Lyon, France
[2]Prysm, Lyon, France

## ABSTRACT

Self-supervised learning for inverse problems allows to train a reconstruction network from noise and/or incomplete data alone. These methods have the potential of enabling learning-based solutions when obtaining ground-truth references for training is expensive or even impossible. In this paper, we propose a new self-supervised learning strategy devised for the challenging setting where measurements are observed via a single incomplete observation model. We introduce a new definition of equivariance in the context of reconstruction networks, and show that the combination of self-supervised splitting losses and equivariant reconstruction networks results in unbiased estimates of the supervised loss. Through a series of experiments on image inpainting, accelerated magnetic resonance imaging, sparse-view computed tomography, and compressive sensing, we demonstrate that the proposed loss achieves state-of-the-art performance in settings with highly rank-deficient forward models.[1]

## 1 INTRODUCTION

Inverse problems are ubiquitous in many sensing and imaging applications. They are written as

$$\boldsymbol{y} = \boldsymbol{A}\boldsymbol{x} + \boldsymbol{\varepsilon} \tag{1}$$

where $\boldsymbol{A} \in \mathbb{R}^{m \times n}$ is the known forward matrix, $\boldsymbol{x} \in \mathbb{R}^n$ is the ground truth image to be estimated, $\boldsymbol{y} \in \mathbb{R}^m$ is the observed measurement vector, and $\boldsymbol{\varepsilon} \in \mathbb{R}^m$ is the unknown noise, generally assumed to follow a Gaussian distribution. This model is suitable for many imaging modalities, including magnetic resonance imaging (MRI) for medical imaging (Zbontar et al., 2019), computed tomography (CT) (Withers et al., 2021), microscopy (Ragone et al., 2023), remote sensing (Fassnacht et al., 2024), and astronomical imaging (Vojtekova et al., 2021).

The number of linearly independent measurements is often smaller than the number of pixels in the target images due to physical and practical constraints. In this case, the forward matrix is rank-deficient, the forward process discards some of the information present in the target image. Further information about the target images is needed to solve the problem and different methods make different assumptions about the signal distribution.

Modern learning-based solvers generally obtain state-of-the-art performance by training on supervised pairs of ground-truth images and measurements. For certain applications, it is expensive or even impossible to obtain enough ground-truth data for supervised training (Belthangady & Royer, 2019). This is notably the case in astronomical imaging, microscopy and medical imaging.

Recent self-supervised methods overcome this limitation by learning to reconstruct without ground-truth data, relying only on a dataset of measurements, and they generally differ in the assumption they make on the forward model. For denoising problems where the forward matrix is the identity mapping, certain methods rely on knowing the exact noise distribution (Eldar, 2009; Pang et al., 2021; Monroy et al., 2025), others only assume it is entry-wise independent (Krull et al., 2019), while some make intermediate assumptions (Tachella et al., 2025a).

In settings where measurements are observed via multiple incomplete forward operators, such as accelerated MRI with masks varying across acquisitions or inpainting problems with missing pixels

---

[*]The first two authors have contributed equally to this paper.
[1]The code is available at `https://github.com/vsechaud/Equivariant-Splitting`

varying across images, the main approaches are splitting (Yaman et al., 2020; Millard & Chiew, 2023) and consistency across operators (Tachella et al., 2022). Splitting losses divide the measurements into input and target components, and are unbiased estimators of the supervised loss if there is enough diversity of operators in the dataset (Daras et al., 2023; Millard & Chiew, 2023).

In the more challenging case of measurement data obtained via a single incomplete forward operator, the main self-supervised approach is equivariant imaging (Chen et al., 2021; 2022) which makes the assumption that the target distribution is invariant to a certain group of transformations including geometric transformations (Wang & Davies, 2024; Scanvic et al., 2025) and range transformations such as intensity scalings (Sechaud et al., 2024). Experimental results show that it can obtain competitive performances to supervised methods even though it does not require ground-truth references for training (Chen et al., 2022). However, training with EI is typically slower than supervised learning, as it requires two to three evaluations of the model for every iteration, and can obtain subpar performances if the operator is highly incomplete.

In this work, we propose equivariant splitting (ES), a new self-supervised method for learning from measurements obtained via a single forward operator that combines the invariance to transformations assumption of equivariant imaging and the simplicity and computational efficiency of splitting methods. The key idea behind our method is the use of recent developments in equivariant architectures (Chaman & Dokmanić, 2021a; Puny et al., 2022) to design a training loss that performs implicit ground truth data augmentation without any transformation overhead. Our theoretical results show that our method yields (in expectation) the minimum mean squared error (MMSE) estimator as long as the model is expressive enough, which is not guaranteed for equivariant imaging and previous splitting methods. Additionally, our experiments demonstrate state-of-the-art performance on a wide range of self-supervised imaging problems including compressed sensing, image inpainting, accelerated MRI and sparse-view CT. This work is additional evidence that architectural constraints built upon equivariance are a powerful tool to solve ill-posed imaging inverse problems.

Our contributions are the following:

1. We propose a new definition for equivariance in inverse problems, and propose architectures that satisfy this property, including unrolled architectures.

2. We propose a new self-supervised loss that leverages equivariant networks (according to our definition above) whose global minimizer is the gold standard MMSE estimator under the assumption of an invariant signal distribution.

3. We demonstrate the performance of our method on a wide range of inverse problems including compressed sensing, inpainting, accelerated MRI and sparse-view computed tomography.

## 2 RELATED WORK

**Measurement splitting** Various self-supervised losses consist of dividing measurement vectors into two components, one used as input and the other as target. It has been used to solve oversampled single-operator inverse problems including full-view CT (Hendriksen et al., 2020), and also undersampled multi-operator inverse problems with theoretical guarantees of producing estimates equivalent to supervised estimates in expectation, notably accelerated MRI and image inpainting with varying masks (Daras et al., 2023; Millard & Chiew, 2023). To the best of our knowledge, this work is the first to extend these methods to the more challenging single-operator undersampled setting, which notably includes sparse-view CT and accelerated MRI (with fixed mask).

**Equivariant imaging** It is possible to learn from incomplete data associated with a single degradation operator, as long as the underlying signal distribution is invariant to a group of transformations (Tachella et al., 2023). Equivariant imaging (Chen et al., 2021; 2022) assumes that the distribution of clean images remains unchanged under certain transformations, including translations, rotations and flips, and introduces a training loss that enforces the equivariance to these transformations of the entire measurement-reconstruction process, thereby constraining the set of learnable models. It has been used effectively to solve various inverse problems using adequate groups of transformations (Wang & Davies, 2024; Sechaud et al., 2024), but it is computationally expensive due to the two to three network evaluations. Moreover, the EI loss is not necessarily an unbiased

estimator of the supervised loss, and it is unclear whether this approach recovers the optimal MMSE estimator in expectation.

**Equivariant neural networks** The design of equivariant networks is an active research topic and many different approaches exist. Some rely on data augmentation to make neural networks more equivariant to rotations and flips, while other rely on averaging on the group of transformations at test time (Rivera et al., 2021; Puny et al., 2022; Kaba et al., 2023; Sannai et al., 2024). A third line of work focuses on architectures that are equivariant by design: Cohen & Welling (2016; 2017) propose convolutional layers that are equivariant to rotations and flips, and other works also rely on the design of more equivariant layers to improve translation-equivariance. Zhang (2019); Chaman & Dokmanić (2021b) propose equivariant pooling and downsampling/upsampling layers, and Karras et al. (2021); Michaeli et al. (2023) equivariant non-linearities and activation layers. Closer to our work, in the specific setting of inverse problems, Celledoni et al. (2021) propose the use of equivariant denoising blocks within unrolled architectures, and Terris et al. (2024) similarly propose to render plug-and-play denoisers more equivariant by averaging over transformations at test time. We take these analyses further by providing a clear definition of equivariance in inverse problems and showing which popular posterior estimators and related architectures verify these properties.

## 3 BACKGROUND

Solving the inverse problem in eq. (1) amounts to designing a reconstruction function $f(\boldsymbol{y}, \boldsymbol{A}) \approx \boldsymbol{x}$ estimating a ground truth signal $\boldsymbol{x} \in \mathbb{R}^n$ from its measurement vector $\boldsymbol{y} \in \mathbb{R}^m$ and forward matrix $\boldsymbol{A} \in \mathbb{R}^{m \times n}$. In practice, it is often implemented as a neural network parametrized by a set of weights. Supervised methods assume the existence of a finite dataset containing pairs of ground truth signals and measurements $\{(\boldsymbol{x}_i, \boldsymbol{y}_i)\}_{i \in \mathcal{I}}$ that are used to learn the reconstruction function $f(\boldsymbol{y}, \boldsymbol{A})$. The main approach is to minimize a training loss equal to the mean squared error (Ongie et al., 2019)

$$\min_f \left\{ \frac{1}{|\mathcal{I}|} \sum_{i \in \mathcal{I}} \mathcal{L}_{\text{SUP}}(\boldsymbol{x}_i, \boldsymbol{y}_i, \boldsymbol{A}, f) \right\}, \quad \mathcal{L}_{\text{SUP}}(\boldsymbol{x}, \boldsymbol{y}, \boldsymbol{A}, f) = \|f(\boldsymbol{y}, \boldsymbol{A}) - \boldsymbol{x}\|^2. \quad (2)$$

While this approach obtains state-of-the-art performance, it cannot be used in the absence of ground-truth data. Self-supervised methods overcome this limitation using a finite dataset containing only measurements $\{\boldsymbol{y}_i\}_{i \in \mathcal{I}}$, and a training loss $\mathcal{L}(\boldsymbol{y}, \boldsymbol{A}, f)$ that need no ground truth data to be evaluated, and which is designed to approximate well the supervised objective in eq. (2).

In denoising problems ($\boldsymbol{A} = \boldsymbol{I}$) with Gaussian noise of known variance, Recorrupted2Recorrupted (R2R) (Pang et al., 2021) and SURE (Metzler et al., 2020) provide unbiased estimators of the supervised loss. The R2R loss is computed by adding synthetic Gaussian noise $\boldsymbol{\omega} \sim \mathcal{N}(\boldsymbol{0}, \sigma^2 \boldsymbol{I})$ to the measurements, creating input-target pairs as $(\boldsymbol{y} + \alpha\boldsymbol{\omega}, \boldsymbol{y} - \frac{\boldsymbol{\omega}}{\alpha})$ for some $\alpha \in (0, +\infty)$. However, if measurements are observed by an incomplete operator $\boldsymbol{A}$ with a non-trivial nullspace, these losses fail to learn in the nullspace, approximating $f(\boldsymbol{y}, \boldsymbol{A}) = \boldsymbol{A}^\dagger \boldsymbol{A} \, \mathbb{E}_{\boldsymbol{x}|\boldsymbol{y}, \boldsymbol{A}} \{\boldsymbol{x}\} + v(\boldsymbol{y}, \boldsymbol{A})$, with $v$ being *any* function taking values in the nullspace of $\boldsymbol{A}$ (Chen et al., 2021).

In the rest of this section, we present the two main self-supervised losses that can learn beyond the nullspace, measurement splitting in Section 3.1 and equivariant imaging in Section 3.2. By combining their main ideas, we obtain our new self-supervised loss introduced in Section 4.

### 3.1 MEASUREMENT SPLITTING

One strategy to address the limitations imposed by the nullspace of a single operator is to employ multiple operators (Millard & Chiew, 2023). The key idea is that different operators generally do not share the same nullspace; thus, observing measurements through the image spaces of multiple operators allows access to the whole space $\mathbb{R}^n$. Formally, they assume that measurements are obtained according to $\boldsymbol{y} \sim p(\boldsymbol{y}|\boldsymbol{A}\boldsymbol{x})$ where the measurement operator $\boldsymbol{A}$ is itself drawn from a distribution $p(\boldsymbol{A})$, and differ for each acquisition.

Splitting losses divide the measurements into two components $\boldsymbol{y} = [\boldsymbol{y}_1^\top, \boldsymbol{y}_2^\top]^\top$ with corresponding operators $\boldsymbol{A} = [\boldsymbol{A}_1^\top, \boldsymbol{A}_2^\top]^\top$, where $\boldsymbol{A}_1 \in \mathbb{R}^{m_1 \times n}$ and $\boldsymbol{A}_2 \in \mathbb{R}^{m_2 \times n}$ with $m_1 + m_2 = m$. The network is then trained to predict the entire measurements vector $\boldsymbol{y}$ from only one of its two

components $\boldsymbol{y}_1$, using the training loss[2]

$$\mathcal{L}_{\text{SPLIT}}(\boldsymbol{y}, \boldsymbol{A}, f) = \mathbb{E}_{\boldsymbol{y}_1, \boldsymbol{A}_1 | \boldsymbol{y}, \boldsymbol{A}} \left\{ \|\boldsymbol{A}f(\boldsymbol{y}_1, \boldsymbol{A}_1) - \boldsymbol{y}\|^2 \right\}, \tag{3}$$

where $p(\boldsymbol{y}_1, \boldsymbol{A}_1 \mid \boldsymbol{y}, \boldsymbol{A})$ is a random splitting distribution, which is chosen on a per-problem basis. The loss encourages the model to learn in the nullspace of each operator by predicting the unobserved part. In practice, the expectation in eq. (3) is estimated using a single split $\boldsymbol{y}_1$ for each training batch.

### 3.2 Equivariant imaging

Equivariant imaging (EI) relies on the assumption that the distribution of images is invariant under a group of transformations $\boldsymbol{T}_g \in \mathbb{R}^{n \times n}$, $g \in \mathcal{G}$, to learn beyond the nullspace of measurements obtained via a *single* operator (Chen et al., 2021; 2022). In this setting, the reconstruction model is expected to be able to estimate ground truth images $\boldsymbol{x}$ as well as their transformations $\boldsymbol{T}_g \boldsymbol{x}$ in a coherent manner, i.e., such that the entire measurement-reconstruction pipeline $f(\boldsymbol{A}\boldsymbol{x})$ is equivariant with respect to the transformations $f(\boldsymbol{A}\boldsymbol{T}_g \boldsymbol{x}, \boldsymbol{A}) = \boldsymbol{T}_g f(\boldsymbol{A}\boldsymbol{x}, \boldsymbol{A})$. In order to achieve this, they propose a self-supervised loss which consists in a traditional measurement consistency term (replaced by SURE in the presence of noise), along with an equivariance-promoting term

$$\mathcal{L}_{\text{EI}}(\boldsymbol{y}, \boldsymbol{A}, f) = \|\boldsymbol{A}f(\boldsymbol{y}, \boldsymbol{A}) - \boldsymbol{y}\|^2 + \lambda \mathbb{E}_g \{\|\boldsymbol{T}_g f(\boldsymbol{y}, \boldsymbol{A}) - f(\boldsymbol{A}\boldsymbol{T}_g f(\boldsymbol{y}, \boldsymbol{A}), \boldsymbol{A})\|^2\}, \tag{4}$$

where $\lambda > 0$ is a trade-off coefficient. Even though it has been shown to be particularly effective on a wide variety of problems (Wang & Davies, 2024; Sechaud et al., 2024), it typically requires from two to three evaluations of the neural network which makes it very computationally expensive (Xu et al., 2025). Moreover, the equivariant loss is only effective at enforcing equivariance when the learned estimator achieves almost perfect reconstructions, $f(\boldsymbol{y}, \boldsymbol{A}) \approx \boldsymbol{x}$ (Chen et al., 2021), which is not the case for very ill-conditioned problems and which leads to the method having multiple possible solutions in general.

## 4 Method

In this section, we present our method that combines measurement splitting and EI introduced in Section 3. In order to learn from incomplete measurements obtained via a single operator, we rely on the same assumption of EI:

**Assumption 1.** *The distribution of ground truth images $p(\boldsymbol{x})$ is invariant to the transformations* $\{\boldsymbol{T}_g\}_{g \in \mathcal{G}}$

$$p(\boldsymbol{T}_g \boldsymbol{x}) = p(\boldsymbol{x}), \ \forall g \in \mathcal{G}, \forall \boldsymbol{x} \in \mathbb{R}^n. \tag{5}$$

The set of transformations is a design choice of the method to be chosen on a per-problem basis. Corollary 1 helps to choose them for specific problems, and we use it in the experiments. This assumption applies in many different settings, natural image distributions are generally invariant to rigid transformations (translations, rotations, flips, scalings), especially microscopic, aerial or remote sensing images which have no privileged orientation. Moreover, our experiments show that our method performs well even in the presence of approximate invariance, e.g., for medical images. Refer to Appendix A.2 for more details about how to choose the transformations for a given application.

In Section 4.1, we present our proposed loss and state its optimality under mild assumptions in Theorem 1 and Proposition 1. In Section 4.2, we present a new definition of equivariance for reconstruction functions and state sufficient conditions for common architectures to be equivariant in Theorem 2. In Section 4.3, we finally present a computational synergy between our loss and these equivariant architectures stated in Theorem 3. See Appendix A.3 for a description of the end-to-end algorithm, and Appendix C for the detailed proofs.

### 4.1 Proposed loss

Under Assumption 1, the measurements can be understood in a different way than they are traditionally. Indeed, measurements $\boldsymbol{y}$ are generally thought of as being associated to the ground truth

---

[2]We use the notation $\mathbb{E}_{a|b}\{g(a)\}$ for $\int_a g(a)p(a|b)\mathrm{d}a$.

image $\boldsymbol{x}$ and the forward matrix $\boldsymbol{A}$, but they can equally be understood as being associated to the virtual ground truth image $\boldsymbol{x}_g = \boldsymbol{T}_g^{-1}\boldsymbol{x}$ and the virtual forward matrix $\boldsymbol{A}_g = \boldsymbol{A}\boldsymbol{T}_g$, with

$$\boldsymbol{y} = \boldsymbol{A}\boldsymbol{x} + \boldsymbol{\varepsilon} = \boldsymbol{A}\boldsymbol{T}_g\boldsymbol{T}_g^{-1}\boldsymbol{x} + \boldsymbol{\varepsilon} = \boldsymbol{A}_g\boldsymbol{x}_g + \boldsymbol{\varepsilon}. \tag{6}$$

The implicit multi-operator structure (with operators $\{\boldsymbol{A}\boldsymbol{T}_g\}_{g\in\mathcal{G}}$) of the problem hints that we should be able to leverage the splitting approaches presented in Section 3. Moreover, since we use the same invariance assumption as EI, we can combine the two approaches to obtain equivariant splitting (ES), a new self-supervised loss $\mathcal{L}_{\text{ES}}$ that has the advantages of both methods.

**Noiseless measurements** The ES self-supervised loss is expressed as

$$\mathcal{L}_{\text{ES}}(\boldsymbol{y}, \boldsymbol{A}, f) \triangleq \mathbb{E}_g\left\{\mathcal{L}_{\text{SPLIT}}(\boldsymbol{y}, \boldsymbol{A}\boldsymbol{T}_g, f)\right\} \tag{7}$$

$$= \mathbb{E}_g\left\{\mathbb{E}_{\boldsymbol{y}_1, \boldsymbol{A}_1|\boldsymbol{y}, \boldsymbol{A}\boldsymbol{T}_g}\left\{\|\boldsymbol{A}\boldsymbol{T}_g f(\boldsymbol{y}_1, \boldsymbol{A}_1) - \boldsymbol{y}\|^2\right\}\right\} \tag{8}$$

where $\boldsymbol{A}_1 \sim p(\boldsymbol{A}_1|\boldsymbol{A}\boldsymbol{T}_g) \triangleq p(\boldsymbol{A}_1|g)$ is a random splitting of $\boldsymbol{A}\boldsymbol{T}_g$.

**Theorem 1.** *In the case of noiseless measurements with $p(\boldsymbol{x})$ $\mathcal{G}$-invariant (Assumption 1), if the matrix $\boldsymbol{Q}_{\boldsymbol{A}_1} \triangleq \mathbb{E}_{g|\boldsymbol{A}_1}\left\{(\boldsymbol{A}\boldsymbol{T}_g)^\top \boldsymbol{A}\boldsymbol{T}_g\right\} \in \mathbb{R}^{n\times n}$ has full rank for some split $\boldsymbol{A}_1$, then the splitting method yields the same MMSE-optimal reconstructions as the supervised method, i.e.,*

$$f^*(\boldsymbol{y}_1, \boldsymbol{A}_1) = \mathbb{E}_{\boldsymbol{x}|\boldsymbol{y}_1, \boldsymbol{A}_1}\{\boldsymbol{x}\}. \tag{9}$$

While it is sufficient for the matrix $\boldsymbol{Q}_{\boldsymbol{A}_1}$ to be invertible for the reconstructions to be almost or exactly optimal, in practice the spectrum (number of its non-negligible eigenvalues) of the matrix determines how close to optimal they are. In the experiments, we use a single (random) split per batch element and we average the reconstructions corresponding to 10 splits at inference.

**Proposition 1.** *If the matrix $\bar{\boldsymbol{Q}}_{\boldsymbol{A}} \triangleq \mathbb{E}_{\boldsymbol{A}_1|\boldsymbol{A}}\{\boldsymbol{Q}_{\boldsymbol{A}_1}\} \in \mathbb{R}^{n\times n}$ is invertible and $f$ minimizes $\mathbb{E}_{\boldsymbol{y}}\{\mathcal{L}_{ES}(\boldsymbol{y}, \boldsymbol{A}, f)\}$. Then the reconstruction function*

$$\overline{f}(\boldsymbol{y}, \boldsymbol{A}) \triangleq \mathbb{E}_{\boldsymbol{y}_1, \boldsymbol{A}_1|\boldsymbol{y}, \boldsymbol{A}}\left\{\bar{\boldsymbol{Q}}_{\boldsymbol{A}}^{-1}\boldsymbol{Q}_{\boldsymbol{A}_1} f(\boldsymbol{y}_1, \boldsymbol{A}_1)\right\} \tag{10}$$

*satisfies*

$$\overline{f}(\boldsymbol{y}, \boldsymbol{A}) = \mathbb{E}_{\boldsymbol{y}_1, \boldsymbol{A}_1|\boldsymbol{y}, \boldsymbol{A}}\left\{\bar{\boldsymbol{Q}}_{\boldsymbol{A}}^{-1}\boldsymbol{Q}_{\boldsymbol{A}_1}\mathbb{E}_{\boldsymbol{x}|\boldsymbol{y}_1, \boldsymbol{A}_1}\{\boldsymbol{x}\}\right\}. \tag{11}$$

*where eq. (11) is a convex combination of MMSE estimators for different splittings.*

In practice, often neither $\bar{\boldsymbol{Q}}_{\boldsymbol{A}}$ nor $\boldsymbol{Q}_{\boldsymbol{A}_1}$ can be computed in closed-form, and we use a non-weighted average over random splittings

$$\overline{f}(\boldsymbol{y}, \boldsymbol{A}) := \frac{1}{J}\sum_{j=1}^{J} f(\boldsymbol{y}_1^{(j)}, \boldsymbol{A}_1^{(j)}) \text{ with } (\boldsymbol{y}_1^{(j)}\boldsymbol{A}_1^{(j)}) \sim p(\boldsymbol{y}_1\boldsymbol{A}_1|\boldsymbol{y}, \boldsymbol{A}\boldsymbol{T}_g) \tag{12}$$

where $g$ is chosen randomly over the group of transformations for each split.

As with EI, the forward operator should not be equivariant with respect to the choice of transformations, in order to learn beyond the nullspace of the operator:

**Corollary 1.** *In order for the matrices $\boldsymbol{Q}_{\boldsymbol{A}_1}$ or $\bar{\boldsymbol{Q}}_{\boldsymbol{A}}$ to have full rank, it is necessary that $\boldsymbol{A}$ is not equivariant:*

$$\exists g \in \mathcal{G}, \boldsymbol{A}\boldsymbol{T}_g \neq \boldsymbol{T}_g\boldsymbol{A}. \tag{13}$$

**Noisy measurements** The ES loss can be split into two separate terms, one enforcing measurement consistency, and the other prediction accuracy:

$$\mathcal{L}_{\text{ES}}(\boldsymbol{y}, \boldsymbol{A}, f) = \mathbb{E}_g\left\{\mathbb{E}_{\boldsymbol{y}_1, \boldsymbol{A}_1|\boldsymbol{y}, \boldsymbol{A}\boldsymbol{T}_g}\left\{\|\boldsymbol{A}_1 f(\boldsymbol{y}_1, \boldsymbol{A}_1) - \boldsymbol{y}_1\|^2 + \|\boldsymbol{A}_2 f(\boldsymbol{y}_1, \boldsymbol{A}_1) - \boldsymbol{y}_2\|^2\right\}\right\},$$

where $\boldsymbol{A}_1$ and $\boldsymbol{A}_2$ are a splitting of $\boldsymbol{A}\boldsymbol{T}_g$. If the measurements are noisy, the first term can be replaced by a self-supervised denoising loss. In particular, if measurements are corrupted by Gaussian noise of standard deviation $\sigma$, we replace the first term by the R2R loss, yielding:

$$\mathcal{L}_{\text{G-ES}}(\boldsymbol{y}, \boldsymbol{A}, f) = \mathbb{E}_{g, \boldsymbol{y}_1, \boldsymbol{A}_1, \boldsymbol{\omega}|\boldsymbol{y}, \boldsymbol{A}\boldsymbol{T}_g}\left\{\left\|\boldsymbol{A}_1 f(\boldsymbol{y}_1 + \alpha\boldsymbol{\omega}, \boldsymbol{A}_1) - \left(\boldsymbol{y}_1 - \frac{\boldsymbol{\omega}}{\alpha}\right)\right\|^2\right.$$

$$\left. + \|\boldsymbol{A}_2 f(\boldsymbol{y}_1 + \alpha\boldsymbol{\omega}, \boldsymbol{A}_1) - \boldsymbol{y}_2\|^2\right\}$$

with $\boldsymbol{\omega} \sim \mathcal{N}(\mathbf{0}, \sigma^2 \boldsymbol{I})$ and a hyper-parameter $\alpha \in (0, +\infty)$. Since R2R provides an unbiased estimate of the clean measurement consistency term (Pang et al., 2021), we can apply Theorem 1 to show that minimizing this loss (in expectation) also results in MMSE estimators (if the conditions on $\boldsymbol{Q}_{\boldsymbol{A}_1}$ or $\bar{\boldsymbol{Q}}_{\boldsymbol{A}}$ are verified). In the case of non-Gaussian noise, the R2R loss can be replaced by its non-Gaussian extension (Monroy et al., 2025). As with random splits, the expectation over $\boldsymbol{\omega}$ is computed using a random realization per batch. At test time, we modify eq. (12) to average over both splits and synthetic noise additions.

## 4.2 EQUIVARIANT RECONSTRUCTORS

The ES loss requires a model evaluation for every mask and transformation. We show that, instead of sampling a random transformation each evaluation, imposing architectural equivariance constraints removes the need to explicitly compute the transforms.

Image-to-image functions $\phi(\boldsymbol{x})$ are equivariant if they satisfy (Cohen & Welling, 2016)

$$\phi(\boldsymbol{T}_g \, \boldsymbol{x}) = \boldsymbol{T}_g \, \phi(\boldsymbol{x}), \ \forall \boldsymbol{x} \in \mathbb{R}^n, \ \forall g \in \mathcal{G}. \tag{14}$$

In this work, we introduce an extension of this definition to reconstruction functions $f(\boldsymbol{y}, \boldsymbol{A})$. To the best of our knowledge, this is the first work that introduces this definition.

**Definition 1.** *We say that the reconstruction function $f(\boldsymbol{y}, \boldsymbol{A})$ is an equivariant reconstructor if*

$$f(\boldsymbol{y}, \boldsymbol{A}\boldsymbol{T}_g) = \boldsymbol{T}_g^{-1} f(\boldsymbol{y}, \boldsymbol{A}), \ \forall \boldsymbol{y} \in \mathbb{R}^m, \forall g \in \mathcal{G}, \forall \boldsymbol{A} \in \mathbb{R}^{m \times n}. \tag{15}$$

This property is very general and the class of classical reconstruction functions that satisfy it is large.

**Theorem 2.** *The reconstruction functions defined in points below are all equivariant as in eq. (15).*

1. **Artifact removal network.** *For a denoiser $\phi(\boldsymbol{x})$ equivariant in the sense of eq. (14),*

$$f(\boldsymbol{y}, \boldsymbol{A}) = \phi\left(\boldsymbol{A}^\top \boldsymbol{y}\right), \ or \ f(\boldsymbol{y}, \boldsymbol{A}) = \phi\left(\boldsymbol{A}^\dagger \boldsymbol{y}\right). \tag{16}$$

2. **Unrolled network.** *For $\phi(\boldsymbol{x})$ equivariant, any $\gamma \in \mathbb{R}$ and data fidelity $d(\boldsymbol{A}\boldsymbol{x}, \boldsymbol{y})$, with*

$$\boldsymbol{x}_0 = \mathbf{0}, \quad \boldsymbol{x}_{k+1} = \phi\left(\boldsymbol{x}_k - \gamma \nabla_{\boldsymbol{x}_k} d(\boldsymbol{A}\boldsymbol{x}_k, \boldsymbol{y})\right) \tag{17}$$

   *for $k = 0, \ldots, L-1$ and $f(\boldsymbol{y}, \boldsymbol{A}) = \boldsymbol{x}_L$.*

3. **Reynolds averaging.** *For a possibly non-equivariant reconstructor $r(\boldsymbol{y}, \boldsymbol{A})$, with*

$$f(\boldsymbol{y}, \boldsymbol{A}) = \frac{1}{|\mathcal{G}|} \sum_{g \in \mathcal{G}} \boldsymbol{T}_g r(\boldsymbol{y}, \boldsymbol{A}\boldsymbol{T}_g). \tag{18}$$

4. **Maximum a posteriori (MAP).** *For a distribution $p(\boldsymbol{x})$ invariant as in eq. (5), with*

$$f(\boldsymbol{y}, \boldsymbol{A}) = \underset{\boldsymbol{x} \in \mathbb{R}^n}{\arg\max} \left\{p(\boldsymbol{x} \mid \boldsymbol{y}, \boldsymbol{A})\right\}. \tag{19}$$

5. **Minimum mean squared error (MMSE).** *For a distribution $p(\boldsymbol{x})$ invariant as in eq. (5),*

$$f(\boldsymbol{y}, \boldsymbol{A}) = \mathbb{E}_{\boldsymbol{x}|\boldsymbol{y}, \boldsymbol{A}} \left\{\boldsymbol{x}\right\}. \tag{20}$$

For additional motivation and details about these reconstructor architectures, see Appendix A.

## 4.3 EFFICIENT LOSS EVALUATION WITH EQUIVARIANT RECONSTRUCTORS

For equivariant reconstructors, the ES loss in eq. (7) reduces to the splitting loss in eq. (3).

**Theorem 3.** *If $f(\boldsymbol{y}, \boldsymbol{A})$ is an equivariant reconstructor, then ES is equivalent to the splitting loss*

$$\mathcal{L}_{\mathrm{ES}}(\boldsymbol{y}, \boldsymbol{A}, f) = \mathcal{L}_{\mathrm{SPLIT}}(\boldsymbol{y}, \boldsymbol{A}, f). \tag{21}$$

We emphasize that the condition for a reconstructor to be equivariant is different from the condition enforced by the EI loss, i.e., $f(\boldsymbol{A}\boldsymbol{T}_g \boldsymbol{x}, \boldsymbol{A}) = \boldsymbol{T}_g f(\boldsymbol{A}\boldsymbol{x}, \boldsymbol{A})$. They are only equivalent if the reconstruction function is a perfect one-to-one mapping over all possible images.

In our experiments, we build equivariant reconstructors using i) artifact removal networks with a translation equivariant UNet denoiser Chaman & Dokmanić (2021b) and ii) unrolled networks with a denoiser architecture equivariant to rotations and flips via averaging (Sannai et al., 2021).

## 5 EXPERIMENTS

We assess the effectiveness of the proposed self-supervised loss using experiments conducted on different inverse problems. For each experiment, we train a model corresponding to our method as well as baseline methods and compare their performance. The inverse problems we consider are 1) inpainting, 2) compressive sensing, 3) accelerated MRI and 4) sparse-view CT. We also validate our theoretical predictions by testing the effect of using an equivariant architecture. In Section 5.1 we detail our experiments on compressive sensing, in Section 5.2 on image inpainting, in Section 5.3 on accelerated MRI and in Section 5.4 on sparse-view CT, and in Section 5.5 we present an ablation study on the effect of equivariant architectures. For additional details about the experiments, see Appendix B.

In each experiment, we compare against multiple baselines: a supervised baseline using the supervised loss described in eq. (2), the EI (Chen et al., 2021) baseline described in eq. (4), a measurement consistency baseline (MC and SURE) Eldar (2009) and a learning-free baseline which are either the measurements directly, or their image under the adjoint or the pseudo-inverse of the forward operator. We use the same architecture and the same training procedure for every method to ensure a fair comparison. We report the peak signal-to-noise ratio (PSNR) and the structural similarity index measure (SSIM) (Wang et al., 2004) of the final reconstructions for the different methods. They are distortion metrics indicating how close the reconstructions are to the ground truth images. We do not include perception metrics which are known to be at odds with them (Blau & Michaeli, 2018). In each case, we also compute equivariance metrics (EQUIV) for translations or rotations and flips.

We design a model architecture from the principles introduced in Section 4, with a variant equivariant to shifts, one equivariant to rotations and flips, and one without equivariance. It uses an existing unrolled architecture (Aggarwal et al., 2019) with a prior step implemented as a standard UNet (Ronneberger et al., 2015), or that of Chaman & Dokmanić (2021b) to enforce the equivariance to shifts. It also optionally uses Reynolds' averaging to enforce the equivariance to rotations and flips. We use a single equivariant variant dictated by Corollary 1 for each problem, the shift-equivariant one for compressive sensing and inpainting, and that equivariant to rotations and flips for MRI and CT.

### 5.1 COMPRESSIVE SENSING

The $28 \times 28$ ground truth images are obtained from the MNIST dataset and are measured without additional noise through compressive matrices $\boldsymbol{A} \in \mathbb{R}^{m \times n}$, with $m < n$ varying from training to training to assess the impact of the compression rate. These matrices are obtained by sampling $A_{i,j} \sim \mathcal{N}(0, 1/m)$ for $i = 1, \ldots, m$ and $j = 1, \ldots, m$, where $n = 28 \times 28$. Figure 1 shows the performance of the different methods for the different compression rates. Our method performs almost as well as the supervised baseline, while the equivariant imaging baseline performs close to the supervised baseline only for higher compression rates.

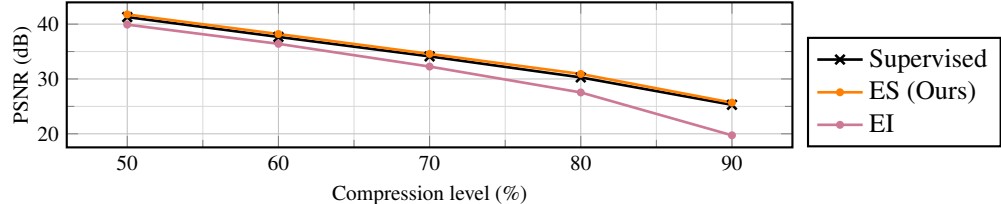

Figure 1: **Compressive sensing results.** ES (ours) performs similarly as the supervised baseline, unlike EI (baseline) whose performance gap widens with higher compression levels.

### 5.2 IMAGE INPAINTING

The dataset consists of $128 \times 128$ images from DIV2K (Agustsson & Timofte, 2017) measured without additional noise through a single subsampling matrix $\boldsymbol{A} \in \mathbb{R}^{m \times n}$ selecting about 30% of the pixels. This matrix is obtained by sampling $a_1, \ldots, a_n \sim \mathcal{B}(0.3)$ where $n = 3 \times 128 \times 128$ and letting $A_{i,j} = \delta_{j_i,j}$ for $i = 1, \ldots, m$ and $j = 1, \ldots, n$, where $m \approx 0.3n$ is the number of nonzero values in $a$, and $j_i$ is the $i$-th index in $a$ corresponding to a nonzero value. Among the 900

images in the dataset, 800 are used for training while the remaining 100 are used for testing. For the supervised method, we use different crops at each evaluation. Table 1 and Figure 2 show that ES performs almost as well as the supervised baseline, and better than EI.

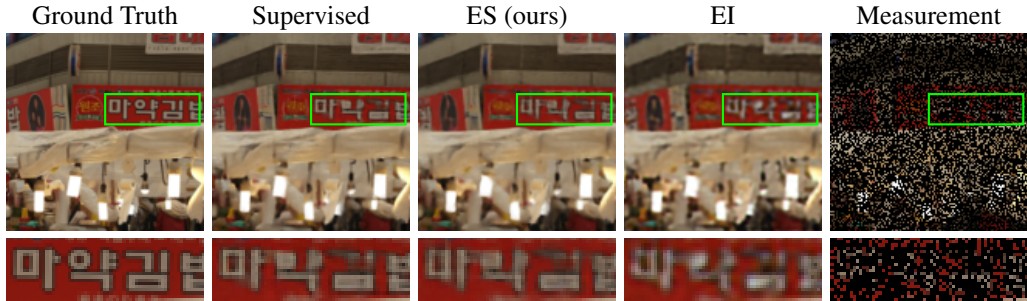

Figure 2: **Sample reconstructions for image inpainting.** ES (ours) produces images perceptually closer to the supervised baseline than EI (baseline) which appears blurry.

Table 1: **Inpainting results.** ES (ours) performs better than EI (baseline), both in terms of reconstruction quality (PSNR, SSIM) and measured equivariance (EQUIV), while performing competitively against the supervised baseline. In **bold**, the best self-supervised metrics (avg $\pm$ st.d.).

| Method | PSNR $\uparrow$ | SSIM $\uparrow$ | EQUIV $\uparrow$ |
|---|---|---|---|
| Supervised | $28.46 \pm 2.97$ | $0.8982 \pm 0.0411$ | $28.46 \pm 2.97$ |
| ES (Ours) | $\mathbf{27.45 \pm 2.86}$ | $\mathbf{0.8737 \pm 0.0461}$ | $\mathbf{27.46 \pm 2.85}$ |
| EI | $25.89 \pm 2.65$ | $0.8332 \pm 0.0521$ | $25.89 \pm 2.65$ |
| MC | $8.22 \pm 2.47$ | $0.0983 \pm 0.0551$ | $8.22 \pm 2.47$ |
| Incomplete image | $8.22 \pm 2.47$ | $0.0973 \pm 0.0542$ | N/A |

## 5.3 Magnetic Resonance Imaging

The dataset contains $320 \times 320$ images from FastMRI (Zbontar et al., 2019) subsampled in the Fourier domain by a single binary mask corresponding to an acceleration of 8, as well as Gaussian noise with a standard deviation of 0.005 corresponding to a signal-to-noise ratio (SNR) of 40 dB. Mathematically, the forward operator $\boldsymbol{A} \in \mathbb{R}^{m \times n}$ is expressed as $\boldsymbol{A} = \mathbf{MF}$ where $\mathbf{F} \in \mathbb{R}^{n \times n}$ denotes the $n \times n$ discrete Fourier transform matrix and where $\mathbf{M} \in \mathbb{R}^{m \times n}$ is the subsampling mask defined as $M_{i,j} = \delta_{j_i,j}$ for $i = 1, \ldots, m$ and $j = 1, \ldots, n$, where $j_i$ denotes the $i$-th component in $\mathbb{R}^n$ corresponding to a pixel in one of the subsampled vertical lines in a random Gaussian mask. Out of the 973 images in the full dataset, 900 are selected for training and the remaining 73 are used for testing. We also test our method on a noise dominated setting using a different mask corresponding to an acceleration of 6, with a higher noise level of 0.1 corresponding to an SNR of only 10 dB, see these results in Appendix B.4. The variant of our proposed loss we use in this experiment is the R2R one introduced in Section 4 with $\alpha = 0.5$. Table 2 shows the performance of the different methods on the test set. Table 3 shows the synergy of our method with the equivariant architecture. Figure 3 shows sample reconstructions from the trained models.

We also evaluate our method on real MRI measurements from FastMRI instead of synthetic measurements. We normalize the k-space data that are sampled on different grids for different scans by resampling them using aliasing-free sinc interpolation on the $320 \times 320$ sampling grid of the ground truth scans. The resulting k-space data are subsampled using the same $\times 8$ acceleration mask used in the setting with synthetic measurements. Table 2 shows that our method performs competitively.

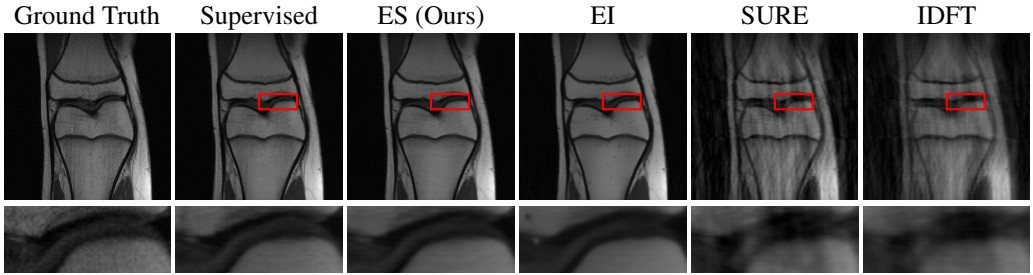

| Ground Truth | Supervised | ES (Ours) | EI | SURE | IDFT |

Figure 3: **Sample reconstructions for MRI (×8 Accel., 40 dB SNR).** Unlike EI (baseline) which suffers from dot-shaped artifacts, ES (ours) is perceptually closer to the supervised baseline. In line with the theoretical predictions, SURE and IDFT (baselines) fail to recover information beyond the observed frequencies.

## 5.4 COMPUTED TOMOGRAPHY

The dataset consists in pairs of ground truth CT scans and corresponding sinograms. In order to obtain the CT scans, we resize to 256 ×256 pixels and clip between -1,000 and 1,000 HUs the ground truth scans from the LIDC-IDRI dataset (Armato III et al., 2011). For each scan, we compute the corresponding sinogram using a discrete Radon transform with 50 views and additive white Gaussian noise with a standard deviation of 0.001 corresponding to a SNR of about 50 dB. The resulting 1,010 pairs are further split into 900 training pairs and 110 test pairs. Table 2 shows that ES performs almost as well as the supervised baseline, and better than EI.

Table 2: **Medical imaging results.** ES (ours) performs better than EI, SURE and MC (baselines), while performing almost as well as the supervised baseline in reconstruction quality (PSNR, SSIM) and measured equivariance (EQUIV). In **bold**, the best self-supervised metrics (avg $\pm$ st.d.).

| Method | PSNR ↑ | SSIM ↑ | EQUIV ↑ |
|---|---|---|---|
| | MRI (×8 Accel., 40 dB SNR) | | |
| Supervised | 28.74 ± 2.81 | 0.6445 ± 0.1094 | 31.71 ± 2.83 |
| ES (Ours) | **28.54 ± 2.75** | **0.6195 ± 0.1188** | **31.53 ± 2.74** |
| EI | 27.88 ± 2.64 | 0.5731 ± 0.1299 | 30.79 ± 2.64 |
| SURE | 24.45 ± 1.86 | 0.5479 ± 0.0740 | 27.35 ± 1.90 |
| IDFT | 23.62 ± 1.90 | 0.5052 ± 0.0900 | 25.99 ± 1.94 |
| | Real MRI measurements (×8 Accel.) | | |
| Supervised | 28.81 ± 2.85 | 0.6480 ± 0.1103 | 31.81 ± 2.84 |
| ES (Ours) | **28.30 ± 2.64** | **0.6151 ± 0.1179** | **31.29 ± 2.62** |
| EI | 27.88 ± 2.61 | 0.5740 ± 0.1290 | 30.80 ± 2.61 |
| MC | 23.63 ± 1.90 | 0.5061 ± 0.0904 | 26.00 ± 1.94 |
| IDFT | 23.63 ± 1.90 | 0.5060 ± 0.0904 | 26.00 ± 1.94 |
| | CT (50 views, 50 dB SNR) | | |
| Supervised | 33.99 ± 2.48 | 0.8819 ± 0.0585 | 34.00 ± 2.49 |
| ES (Ours) | **32.62 ± 2.16** | **0.8570 ± 0.0596** | **32.60 ± 2.17** |
| EI | 28.61 ± 1.28 | 0.7400 ± 0.0466 | 28.61 ± 1.29 |
| FBP | 25.59 ± 0.69 | 0.4805 ± 0.0363 | 25.59 ± 0.70 |

## 5.5 ABLATION STUDY

Table 3 shows that architectures designed to be equivariant are measurably more equivariant across imaging modalities and training losses. Moreover, it shows that networks trained using the splitting loss perform better for equivariant architectures than for non-equivariant architectures, with an even greater gap than for supervised inpainting baselines, which confirms the theoretical analysis made in Section 4. Finally, we observe that non-equivariant architectures still lead to fairly measurably equivariant models. While surprising, this phenomenon has already been witnessed and is usually referred to as learned equivariance, whereby the training data and inductive biases lead to fairly equivariant learned models (Gruver et al., 2024). We believe that this learned equivariance is responsible for the high performance of splitting methods even when using non-equivariant architectures. For additional results on the impact of equivariant architectures, see Appendix B.4.

Table 3: **Impact of using equivariant architectures.** In accordance with the theoretical results described in Section 4, there is a synergy between the splitting loss and equivariant architectures resulting in higher performance. Non-equivariant models have surprisingly high equivariance measures (EQUIV) which might explain their high performance when using the splitting loss. Eq. arch. denotes whether the architecture is equivariant. In **bold**, the best self-supervised metrics (avg $\pm$ st.d.).

| Training loss | Eq. arch. | Image inpainting | | |
| --- | --- | --- | --- | --- |
| | | PSNR $\uparrow$ | SSIM $\uparrow$ | EQUIV $\uparrow$ |
| Supervised | ✓ | $28.46 \pm 2.97$ | $0.8982 \pm 0.0411$ | $28.46 \pm 2.97$ |
| | ✗ | $28.62 \pm 3.03$ | $0.9002 \pm 0.0414$ | $27.85 \pm 2.71$ |
| Splitting (Ours) | ✓ | $\mathbf{27.45 \pm 2.86}$ | $\mathbf{0.8737 \pm 0.0461}$ | $\mathbf{27.46 \pm 2.85}$ |
| | ✗ | $27.20 \pm 2.83$ | $0.8652 \pm 0.0461$ | $26.52 \pm 2.60$ |

| Training loss | Eq. arch. | MRI ($\times 8$ Accel., 40 dB SNR) | | |
| --- | --- | --- | --- | --- |
| | | PSNR $\uparrow$ | SSIM $\uparrow$ | EQUIV $\uparrow$ |
| Supervised | ✓ | $28.74 \pm 2.81$ | $0.6445 \pm 0.1094$ | $31.71 \pm 2.83$ |
| | ✗ | $28.48 \pm 2.68$ | $0.6381 \pm 0.1082$ | $28.78 \pm 1.95$ |
| Splitting (Ours) | ✓ | $\mathbf{28.54 \pm 2.75}$ | $\mathbf{0.6195 \pm 0.1188}$ | $\mathbf{31.53 \pm 2.74}$ |
| | ✗ | $28.18 \pm 2.58$ | $0.6104 \pm 0.1176$ | $27.28 \pm 2.10$ |

## 6 CONCLUSION

In this work, we propose a new self-supervised loss for solving inverse problems which bridges the gap between existing equivariance and splitting-based self-supervised losses. We motivate the design of our loss by showing that minimizing the expected loss results in MMSE estimators. We further validate our method using numerical simulations on different image distributions and imaging modalities, including inpainting of natural images, MRI and CT. These results suggest that the proposed method compares favorably to the equivariant imaging baseline and is close to supervised methods. To the best of our knowledge, this work is the first to leverage equivariant networks to learn from incomplete data alone, going beyond the usual goal of improving the generalization of the networks to unseen transformations at test time. Our method provides a new way to evaluate the benefits of using different equivariant architectures, and can benefit from future advances made in this field. More broadly, our work is further evidence that invariance is a promising prior for learning from incomplete data.

## 7 ACKNOWLEDGEMENTS

Victor Sechaud and Julian Tachella are supported by the ANR grant UNLIP (ANR-23-CE23-0013). This project was provided with computing HPC and storage resources by GENCI at IDRIS thanks to the grant 2025-AD011016422 on the supercomputer Jean Zay's H100 partition. Part of the computations were performed on the machine cluster at the Pascal Blaise Center at the ENS de Lyon (Quemener & Corvellec, 2013).

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

## A  DETAILS ABOUT THE METHOD

### A.1  EQUIVARIANT RECONSTRUCTOR

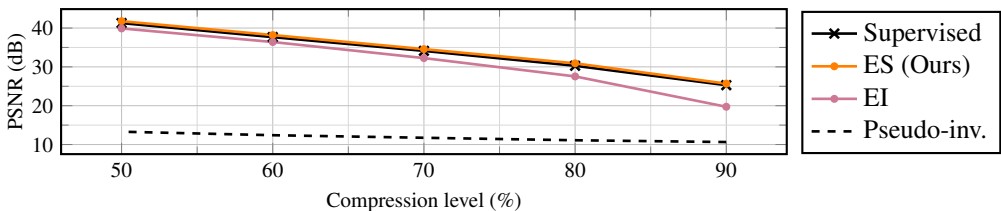

Figure 4: **Compressive sensing results.** Adds the pseudo-inverse reconstruction (pseudo-inv.) as a baseline to Figure 1

Equivariant reconstructor

$$\boldsymbol{y} \longrightarrow \frac{1}{|\mathcal{G}|}\sum_{g\in\mathcal{G}} \boldsymbol{T}_g\, r(\boldsymbol{y}, \boldsymbol{A}\boldsymbol{T}_g) \longrightarrow f(\boldsymbol{y}, \boldsymbol{A})$$

Figure 5: **Reconstructor equivariant to rotations and flips.** It is the Reynolds averaging of 90° rotations and horizontal and vertical flips as in eq. (18) of a non-equivariant reconstructor of the MAP type, as in eq. (19), implemented using a MoDL unrolled algorithm (Aggarwal et al., 2019) with 3 iterations, shared weights, and using a non-equivariant residual UNet (Ronneberger et al., 2015) as the denoiser architecture.

In Theorem 2, we consider reconstruction functions of 4 different structures. In this section, we give additional details about them.

The artifact removal reconstructor architecture (Jin et al., 2017) in eq. (16) consists in a projection step that maps the measurements back into the image space using the adjoint or the pseudo-inverse

of the forward matrix, which is immediately followed by a very general trainable network, often a UNet or another encoder-decoder type of network, or sometimes simply a fully convolutional network. Theorem 2 states that this reconstructor architecture produces equivariant reconstructors as long as the denoiser architecture is, itself, equivariant in the sense of eq. (14).

Reynolds averaging for possibly non-equivariant reconstruction functions $r(\boldsymbol{y}, \boldsymbol{A})$ in eq. (18)

$$f(\boldsymbol{y}, \boldsymbol{A}) = \frac{1}{|\mathcal{G}|} \sum_{g \in \mathcal{G}} \boldsymbol{T}_g r(\boldsymbol{y}, \boldsymbol{A}\boldsymbol{T}_g). \tag{18}$$

has, as far as we know, not been defined in previous works. It is a natural extension of Reynolds averaging for possibly non-equivariant image-to-image functions or denoisers $\psi(\boldsymbol{x})$ (Sannai et al., 2024; Terris et al., 2024),

$$\phi(\boldsymbol{x}) = \frac{1}{|\mathcal{G}|} \sum_{g \in \mathcal{G}} \boldsymbol{T}_g^{-1} \psi(\boldsymbol{T}_g \boldsymbol{x}) \tag{22}$$

which makes them equivariant in the sense of eq. (14), to reconstruction functions in order to make them equivariant in the sense of eq. (15). It is a fairly simple way to make a reconstructor equivariant and it is relatively inexpensive for small groups of transformations such as the group of 90° rotations and horizontal and vertical flips (Cohen & Welling, 2016). It is however too expensive to be used in practice when the group is relatively large, like the group of shifts, since it would require to evaluate the neural network for as many times as there are pixels in the input image, for each image.

The maximum a posteriori (MAP) reconstructor architecture defined in eq. (19)

$$f(\boldsymbol{y}, \boldsymbol{A}) = \underset{\boldsymbol{x} \in \mathbb{R}^n}{\mathrm{argmax}} \left\{ p(\boldsymbol{x} \mid \boldsymbol{y}, \boldsymbol{A}) \right\}. \tag{19}$$

is a classical reconstructor architecture that has been used in different ways, including iterative algorithms with a hand-crafted prior (Rudin et al., 1992; Davy et al., 2025), plug-and-play architectures using an iterative approach but with the hand-crafted prior replaced with a pre-trained denoiser (Venkatakrishnan et al., 2013), and unrolled architectures where the optimization problem is done with a fixed number of steps and where parts of the algorithm are replaced with trainable modules (Aggarwal et al., 2019). It is often interpreted as a Bayesian maximum a posteriori estimator, but also commonly outside of a Bayesian framework as a variational approach with a data-fidelity term and a regularization term. It is one of the reconstructor architectures that we use for our experiments in Section 5. Theorem 2 states that these reconstructors are equivariant as long as the prior distribution is invariant in the sense of eq. (5), or equivalently, if the associated regularization function or negative log-prior is invariant.

The minimum mean squared error (MMSE) estimator in eq. (20)

$$f(\boldsymbol{y}, \boldsymbol{A}) = \mathbb{E}_{\boldsymbol{x} \mid \boldsymbol{y}, \boldsymbol{A}} \left\{ \boldsymbol{x} \right\} \tag{20}$$

is generally the target theoretical reconstructor as it achieves the highest theoretical PSNR (Tachella et al., 2023). It is generally estimated by reconstructors trained with a mean squared error loss (Chen et al., 2021), which is notably how we train the supervised baselines in the experiments in Section 5. Theorem 2 states that as long as the prior distribution is invariant in the sense of eq. (5), the MMSE reconstructor is equivariant in the sense of eq. (15).

### A.2 HOW TO CHOOSE TRANSFORMATIONS FOR A GIVEN INVERSE PROBLEM

There are two major criteria for choosing the transformations for a given application. First, the image distribution of interest should be invariant to the chosen transformations. Aerial, remote sensing and microscopic images are invariant to translations and rotations as the scenes and subjects they measure exhibit no privileged position and orientation with respect to the image plane. Natural image distributions and texture distributions (Portilla & Simoncelli, 2000) are also generally invariant to translations but they are less invariant to rotations as natural images are typically oriented upward and texture distributions might be anisotropic.

Second, the transformations should also be chosen in accordance with the measurement operator. Corollary 1 shows that transformations for which the operator is equivariant do not improve the

reconstruction process. It is a well-known criterion introduced in the original work on equivariant imaging (Chen et al., 2021) which remains correct in our setting where measurement splitting is added to the theoretical analysis. Table 4 lists correct choices of transformations for common measurement operators.

Table 4: **Decision table for the transformations.** Corollary 1 shows that not all transformations are well-suited for all problems. Namely, transformations for which the operator is equivariant introduce no additional information and should not be used. This table specifies which transformations are well-suited for which operator.

| Operator | Translation | Rotation | Permutation | Amplitude |
|---|---|---|---|---|
| Isotropic blur | ✗ | ✗ | ✓ | ✗ |
| Image inpainting | ✓ | ✓ | ✓ | ✗ |
| Sparse-view CT | ✗ | ✓ | ✓ | ✗ |
| Accelerated MRI | ✗ | ✓ | ✓ | ✗ |
| Compressive sensing | ✓ | ✓ | ✓ | ✗ |

### A.3 END-TO-END ALGORITHM

In this section, we present the end-to-end ES algorithm. It consists in a training step where an equivariant reconstructor is trained using backpropagation against a training dataset of measurements only, after which it is applied to obtain the reconstructions associated to the test measurements. The ES loss used in the training step is computed using the expression in Equation (3). The expectations are estimated using Monte Carlo sampling where a single sample is used at training time and $T = 10$ samples are used at inference. In the experiments, we use the splitting ratio $s = 0.8$ corresponding to $m_1 = 0.8m$ and $m_2 = 0.2m$. Algorithms 1 and 2 show the detailed algorithms in pseudo-code.

---

**Algorithm 1:** Equivariant Splitting (Training procedure)

---

**Input:** Dataset $D = \{\boldsymbol{y}_i\}_{i \in I}$, forward operator $\boldsymbol{A} \in \mathbb{R}^{m \times n}$, split ratio $s$, learning rate $\eta$,
        number of epochs $E$, equivariant reconstructor $f_{\boldsymbol{\theta}}$
**Output:** Trained model $f_{\boldsymbol{\theta}}$
**for** *epoch* = 1 **to** $E$ **do**
    Shuffle dataset $D$ ;
    **foreach** *mini-batch* $B \subset D$ **do**
        **init:** $L = 0$
        **foreach** $\boldsymbol{y}$ *in* $B$ **do**
            **init:** $\boldsymbol{M} \in \mathbb{R}^{m_1 \times m}$ a random split s.t $m_1 = s \times m$.
            Split measurement and operator: $\boldsymbol{y}_1 = \boldsymbol{M}\boldsymbol{y}, \boldsymbol{A}_1 = \boldsymbol{M}\boldsymbol{A}$ ;
            Compute predictions: $\hat{\boldsymbol{x}} = f_{\boldsymbol{\theta}}(\boldsymbol{y}_1, \boldsymbol{A}_1)$ ;
            Compute sample loss $\ell = \|\boldsymbol{A}\hat{\boldsymbol{x}} - \boldsymbol{y}\|_2^2$ ;
            Aggregate loss: $L \leftarrow L + \ell$ ;
        **end**
        Mean: $L \leftarrow \frac{1}{|B|}L$ ;
        Backpropagation: compute gradients $\nabla_{\boldsymbol{\theta}} L$ ;
        Update parameters: $\boldsymbol{\theta} \leftarrow \texttt{ADAM}(\boldsymbol{\theta}, \nabla_{\boldsymbol{\theta}} L, \eta)$ ;
    **end**
**end**

---

## B DETAILS ABOUT THE EXPERIMENTS

We use the optimizer AdamW (Loshchilov & Hutter, 2019) for every training with different learning rates for the different inverse problems, a weight decay of $10^{-8}$, beta coefficients equal to 0.9 and

---

**Algorithm 2:** Equivariant Splitting (Inference)

---

**Input:** Measurement $y \in \mathbb{R}^{m \times n}$, forward operator $A \in \mathbb{R}^{m \times n}$, split ratio $s$, number of
      samples $T$, trained reconstructor $f_{\boldsymbol{\theta}}$
**Output:** Reconstructed images $\hat{x} = \overline{f}_{\boldsymbol{\theta}}(y, A)$
**init:** $\hat{x} = 0$
**for** $i = 1$ **to** $T$ **do**
      **init:** $M \in \mathbb{R}^{m_1 \times m}$ a random split s.t $m_1 = s \times m$.
      Split measurement and operator: $y_1 = My, A_1 = MA$ ;
      Compute one predictions: $\hat{x}_i = f_{\boldsymbol{\theta}}(y_1, A_1)$ ;
      Aggregate: $\hat{x} \leftarrow \hat{x} + \hat{x}_i$ ;
**end**
Mean: $\hat{x} \leftarrow \frac{1}{T}\hat{x}$ ;

---

0.999 and without the AMSGrad option. For longer trainings, we use step schedulers that divide the learning rate by a factor ranging from 2 to 10 at specific epochs, up to 3 or 4 times.

In our experiments, we make extensive use of the DeepInverse library (Tachella et al., 2025b) that provides an implementation of the various forward operators and training losses that we use. Every model is trained for up to 50 hours on a single GPU, either an NVIDIA H100, GH200 or RTX 4090. See Appendix B.4 for more details about the training durations.

×8 Accel., 40 dB SNR       ×6 Accel., 10 dB SNR

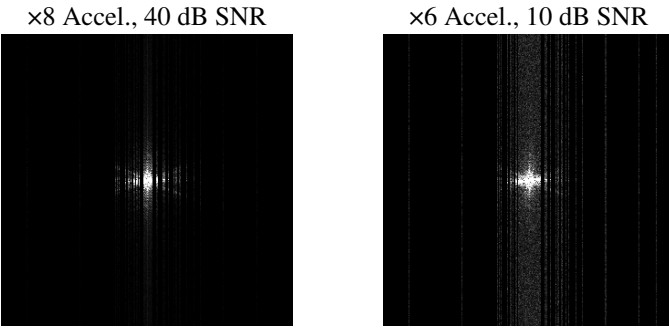

Figure 6: **Examples of k-space measurements in MRI.** (left) a less noisy problem with less measurements, (right) a noisier problem with more measurements.

## B.1 DATASETS

**Image inpainting** The ground truth images are images obtained from the dataset DIV2K (Agustsson & Timofte, 2017) which contains pictures of natural scenes (landscapes, animals) by first resizing them to a resolution of $256 \times 256$ pixels before extracting a central $128 \times 128$ pixels crop. We synthesize the measurements by corrupting the ground truth images using a single binary mask sampled from a pixel-wise Bernoulli distribution with a 30% chance of keeping each pixel value. In this setting, we do not corrupt the measurements further with additional noise.

**MRI** The dataset consists in 973 pairs of ground truth images from the FastMRI dataset (Zbontar et al., 2019) and associated k-space measurements synthesized using a single coil sensitivity map. The ground truth images have a size of $320 \times 320$ pixels and each corresponds to the middle slice of a different 3d knee acquisition. The k-spaces are synthesized as discrete Fourier transforms subsampled on a single non-regular grid and further corrupted with additive white Gaussian noise with a standard deviation of 0.005 corresponding to an average SNR of about 40 dB. The subsampling grid models the coil sensitivity map, is sampled from a Gaussian distribution and corresponds to an acceleration of 8. The entire dataset is finally split into a train/val split containing 900 images and a test split containing the remaining 73 images. Our implementation uses the work from Wang et al. (2025).

We also consider an additional dataset obtained in the same way except using a different mask corresponding to an acceleration of 6 and with a higher noise level corresponding to a standard deviation of 0.1 or an average SNR of 10 dB in the k-space domain.

## B.2 NETWORK ARCHITECTURES

The unrolled architecture uses 3 iterations and the weights are shared across different iterations. Every UNet is residual, has 4 scales and has no normalization layer as we find them to be detrimental to the performance.

The transforms we use in the model architecture and in the metrics are grid-preserving: grid-aligned shifts, 90° rotations, and vertical and horizontal flips. The transforms we use in the EI are grid-aligned shifts and 1° rotations following the prior art Chen et al. (2021). Reynolds' averaging is implemented using an unbiased Monte-Carlo estimator whereby a single random transform is sampled at every evaluation to save on computational cost.

## B.3 METRICS

In the experiments, we use three different performance metrics including two standard distortion metrics (PSNR, SSIM) and a new equivariance metric for reconstructors (EQUIV). The peak signal-to-noise ratio is defined, for images with a dynamic range normalized to $[0, 1]$, as the mean squared error expressed in decibels (dB)

$$\text{PSNR} = \mathbb{E}_{\boldsymbol{x},\boldsymbol{y}} \left\{ -10 \log_{10} \left( \|f(\boldsymbol{y}, \boldsymbol{A}) - \boldsymbol{x}\|^2 \right) \right\}. \tag{23}$$

The structural similarity index measure (Wang et al., 2004) is a more perceptual metric which is a combination of the empirical means, standard deviations and correlation coefficient associated to the reference and compared images. The complete definition of the metric is too long to be included in this work and we refer the reader to the original publication for more details. In addition to these two standard distortion metrics, we use a new equivariance metric similar to that used by Chaman & Dokmanić (2021b) adapted for our proposed definition of equivariant reconstructors. It is the average mean squared error associated with eq. (15) and expressed in dB for readability

$$\text{EQUIV} = -10 \log_{10} \left( \mathbb{E}_{\boldsymbol{y},g} \left\{ \left\| f(\boldsymbol{y}, \boldsymbol{A}\boldsymbol{T}_g) - \boldsymbol{T}_g^{-1} f(\boldsymbol{y}, \boldsymbol{A}) \right\|^2 \right\} \right). \tag{24}$$

It can be roughly understood as a PSNR for equivariance. In every experiment, we use the same group of transformations for EQUIV as we use for the equivariant reconstructor architectures.

## B.4 RESULTS

Table 5: **MRI results in another setting.** Supplementary results for a different MRI problem (×6 Accel., 10 dB SNR) than in Table 2 In **bold**, the best self-supervised metrics. Values: avg $\pm$ st.d.

| Method | MRI (×6 Accel., 10 dB SNR) | | |
| --- | --- | --- | --- |
| | PSNR ↑ | SSIM ↑ | EQUIV ↑ |
| Supervised | 27.39 ± 2.44 | 0.5243 ± 0.1373 | 30.38 ± 2.43 |
| ES (Ours) | **27.33 ± 2.45** | **0.5126 ± 0.1444** | **30.32 ± 2.44** |
| EI | 27.23 ± 2.41 | 0.5110 ± 0.1421 | 30.21 ± 2.40 |
| SURE | 27.08 ± 2.29 | 0.5097 ± 0.1372 | 30.06 ± 2.28 |
| IDFT | 23.85 ± 1.05 | 0.3878 ± 0.0272 | 25.14 ± 0.79 |

In this section, we provide more details about the main experiments and present additional experiments.

For each training done in the main experiments, we report the average epoch duration and the number of epochs until the model has finished training. For the sake of comparability, we made sure to conduct the different trainings for the same imaging modality on the same GPU. Namely, we used a single NVIDIA RTX 3090 Ti GPU for every inpainting experiment and a single NVIDIA H100

GPU for every MRI experiment. Table 10 shows that self-supervised methods including ours have generally longer epochs and require more epochs than the fully supervised gold standard. We believe that this is due to a fundamental trade-off whereby the use of ground truth data accelerates the learning procedure while also enabling the use of simpler training algorithms. Moreover, unlike standard implementations of EI and SURE, ES requires only a single network pass per iteration resulting in significantly faster epochs. Overall, ES is computationally more efficient than EI, in addition to being more performant in terms of reconstruction quality.

Figure 4 shows additional compressive sensing results, Table 5 and Figure 7 show results on a MRI experiment with settings different from the main one, Table 6 shows extended results for the ablation study on equivariant architectures, and Figure 6 shows sample k-spaces from the MRI experiments. Figure 5 shows the equivariant reconstructor architecture that we adopt in the MRI experiments.

Table 6: **Extended results on the impact of equivariant architectures.** Adds to Table 3 the results for EI with a non-equivariant architecture for the inpainting task, results for the noise-dominated MRI task. In **bold**, the best self-supervised metrics. Values: avg $\pm$ st.d.

| Training loss | Eq. arch. | Image inpainting | | |
| --- | --- | --- | --- | --- |
| | | PSNR $\uparrow$ | SSIM $\uparrow$ | EQUIV $\uparrow$ |
| Supervised | ✓ | $28.46 \pm 2.97$ | $0.8982 \pm 0.0411$ | $28.46 \pm 2.97$ |
| | ✗ | $28.62 \pm 3.03$ | $0.9001 \pm 0.0415$ | $27.85 \pm 2.71$ |
| Splitting (Ours) | ✓ | $\mathbf{27.45 \pm 2.86}$ | $0.8737 \pm 0.0461$ | $27.46 \pm 2.85$ |
| | ✗ | $27.20 \pm 2.83$ | $0.8651 \pm 0.0463$ | $26.52 \pm 2.60$ |
| EI loss | ✓ | $25.89 \pm 2.65$ | $0.8332 \pm 0.0521$ | $25.89 \pm 2.65$ |
| | ✗ | $26.33 \pm 2.81$ | $0.8451 \pm 0.0536$ | $25.58 \pm 2.52$ |
| MC loss | ✓ | $8.22 \pm 2.47$ | $0.098 \pm 0.055$ | $8.22 \pm 2.47$ |
| | ✗ | $8.24 \pm 2.48$ | $0.100 \pm 0.056$ | $8.24 \pm 2.48$ |

| Training loss | Eq. arch. | MRI (×8 Accel., 40 dB SNR) | | |
| --- | --- | --- | --- | --- |
| | | PSNR $\uparrow$ | SSIM $\uparrow$ | EQUIV $\uparrow$ |
| Supervised | ✓ | $28.74 \pm 2.81$ | $0.6445 \pm 0.1094$ | $31.71 \pm 2.83$ |
| | ✗ | $28.48 \pm 2.68$ | $0.6381 \pm 0.1082$ | $28.78 \pm 1.95$ |
| Splitting (Ours) | ✓ | $\mathbf{28.54 \pm 2.75}$ | $\mathbf{0.6195 \pm 0.1188}$ | $\mathbf{31.53 \pm 2.74}$ |
| | ✗ | $28.18 \pm 2.58$ | $0.6104 \pm 0.1176$ | $27.28 \pm 2.10$ |

| Training loss | Eq. arch. | MRI (×6 Accel., 10 dB SNR) | | |
| --- | --- | --- | --- | --- |
| | | PSNR $\uparrow$ | SSIM $\uparrow$ | EQUIV $\uparrow$ |
| Supervised | ✓ | $27.39 \pm 2.44$ | $0.5243 \pm 0.1373$ | $30.38 \pm 2.43$ |
| | ✗ | $27.33 \pm 2.42$ | $0.5174 \pm 0.1410$ | $29.73 \pm 2.20$ |
| Splitting (Ours) | ✓ | $\mathbf{27.33 \pm 2.45}$ | $\mathbf{0.5126 \pm 0.1444}$ | $\mathbf{30.32 \pm 2.44}$ |
| | ✗ | $27.20 \pm 2.38$ | $0.5095 \pm 0.1430$ | $28.66 \pm 1.76$ |

We present an additional inpainting experiment, designed to simulate a more realistic acquisition scenario and study the consistency of our method. Specifically, we consider a noisy inpainting setting in which entire image columns are randomly removed. Such sampling patterns naturally arise in satellite imaging systems based on push-broom scanners (PBS) (Xu et al., 2016). As reported in Table 7, the proposed method achieves performance comparable to that of the EI baseline, confirming its consistency under this more practical acquisition model.

To clarify the behavior of ES when the forward operator $A$ is equivariant or nearly equivariant, we conducted the following experiment: we performed an PBS inpainting operation in which every other column of the image is removed. In this setting, the operator $A$ becomes "almost" equivariant in the sense that for any even horizontal shift $g$, we have $AT_g = T_gA$, and the same property holds for all vertical shifts. Table 7 shows the reconstruction performance deteriorates significantly.

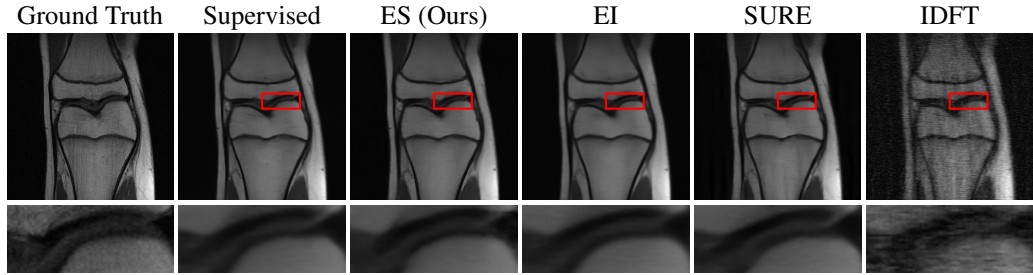

Figure 7: **Sample reconstructions for noise dominated MRI (×6 Accel., 10 dB SNR).** In the noise dominated setting, the different models perform more similarly than in the less noisy setting shown in Figure 3.

Table 7: **Inpainting with push-broom masks.** We consider an unevenly-spaced mask sampled randomly to test our method in a realistic inpainting setting, and an evenly-spaced mask to verify the claim in Corollary 1 empirically in a case of almost-equivariance.

| Method | Image inpainting (Randomly-spaced push-broom mask) | |
| --- | --- | --- |
| | PSNR ↑ | SSIM ↑ |
| Supervised | 23.72 ± 2.10 | 0.743 ± 0.051 |
| ES (Ours) | 23.04 ± 1.81 | 0.734 ± 0.050 |
| EI | 23.05 ± 2.13 | 0.707 ± 0.066 |
| Incomplete image | 9.44 ± 2.31 | 0.141 ± 0.048 |
| Method | Image inpainting (Evenly-spaced push-broom mask) | |
| | PSNR ↑ | SSIM ↑ |
| Supervised | 28.37 ± 2.15 | 0.873 ± 0.035 |
| ES (Ours) | 21.94 ± 2.12 | 0.617 ± 0.091 |
| Incomplete image | 9.61 ± 2.42 | 0.152 ± 0.061 |

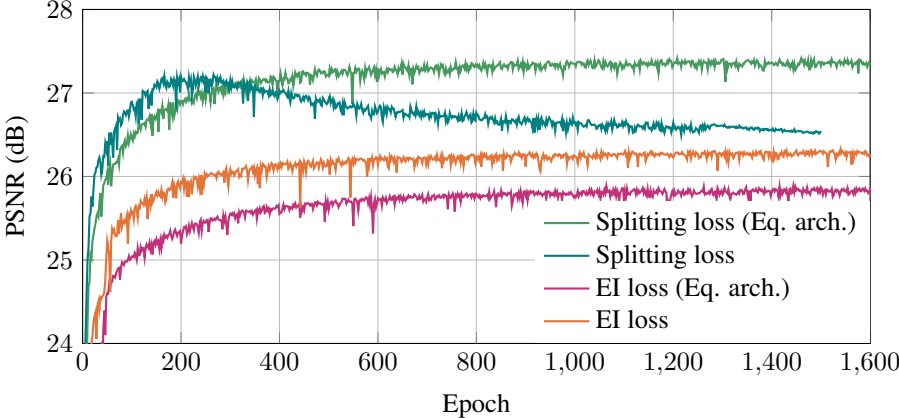

Figure 8: **Performance evolution during training for inpainting.** Splitting methods perform better than EI independent of the network architecture.

### B.4.1 TESTING THE EFFECT OF UNKNOWN NOISE DISTRIBUTIONS

In addition to the main experiments where we use the knowledge of the noise distribution in the reconstruction algorithm, we test a more realistic scenario where the noise distribution is unknown and is estimated to be zero for a lack of a better estimate. To do so, we train models using variants of ES and EI where the measurements are assumed to be noiseless even though the training and testing data are nonetheless corrupted with noise. Table 8 shows that ES also performs better than EI when the noise distribution is unknown. Moreover, it shows that the performance of ES tends to be lower when the noise distribution is unknown, but it does not drop exceedingly which demonstrates the stability of ES to unexpected noise.

Table 8: **Performance when the noise distribution is unknown.** In the unknown noise scenarios, we use the variants of ES and EI corresponding to assuming that the measurements are noiseless.

| Method | Known | Image inpainting | |
| --- | --- | --- | --- |
| | | PSNR $\uparrow$ | SSIM $\uparrow$ |
| Supervised | | $23.72 \pm 2.10$ | $0.743 \pm 0.051$ |
| ES (Ours) | ✓ | $23.04 \pm 1.81$ | $0.734 \pm 0.050$ |
| ES | ✗ | $22.05 \pm 1.36$ | $0.59 \pm 0.061$ |
| EI | ✓ | $23.05 \pm 2.13$ | $0.707 \pm 0.066$ |
| EI | ✗ | $22.01 \pm 1.71$ | $0.59 \pm 0.051$ |
| Incomplete image | | $9.44 \pm 2.31$ | $0.141 \pm 0.048$ |
| Method | Known | MRI (×8 Accel., 40 dB SNR) | |
| | | PSNR $\uparrow$ | SSIM $\uparrow$ |
| Supervised | | $28.74 \pm 2.81$ | $0.6445 \pm 0.1094$ |
| ES (Ours) | ✓ | $28.54 \pm 2.75$ | $0.6195 \pm 0.1188$ |
| ES | ✗ | $28.52 \pm 2.75$ | $0.6194 \pm 0.1191$ |
| EI | ✓ | $27.88 \pm 2.64$ | $0.5731 \pm 0.1299$ |
| EI | ✗ | $27.89 \pm 2.59$ | $0.5755 \pm 0.1285$ |
| IDFT | | $23.62 \pm 1.90$ | $0.5052 \pm 0.0900$ |
| Method | Known | MRI (×6 Accel., 10 dB SNR) | |
| | | PSNR $\uparrow$ | SSIM $\uparrow$ |
| Supervised | | $27.39 \pm 2.44$ | $0.5243 \pm 0.1373$ |
| ES (Ours) | ✓ | $27.33 \pm 2.45$ | $0.5126 \pm 0.1444$ |
| ES | ✗ | $25.73 \pm 1.49$ | $0.4566 \pm 0.0622$ |
| EI | ✓ | $27.23 \pm 2.41$ | $0.5110 \pm 0.1421$ |
| EI | ✗ | $26.02 \pm 1.65$ | $0.4706 \pm 0.0873$ |
| IDFT | | $23.85 \pm 1.05$ | $0.3878 \pm 0.0272$ |

### B.4.2 EMPIRICAL VERIFICATION OF THE EQUIVARIANCE OF MAP RECONSTRUCTORS

We verify empirically that MAP reconstructors are equivariant as long as the prior is itself equivariant, i.e., the claim made in Theorem 2. Since they cannot be computed exactly in general, we consider a specific scenario where they can. We assume that 1) $x \sim \mathcal{N}(0, \tau^2 \boldsymbol{I}_n)$, 2) $A \in \mathbb{R}^{m \times n}$ is the two-dimensional decimation operator with decimation rate 2, 3) $y \mid Ax \sim \mathcal{N}(0, \sigma^2 \boldsymbol{I}_m)$, and 4) that $\boldsymbol{T}_g$ denotes the rotation by angle $g \in \{0°, 90°, 180°, 270°\}$. Under these assumptions, the prior distribution is equivariant and the MAP estimator in eq. (19) can be expressed in closed-form as

$$f(\boldsymbol{y}, \boldsymbol{A}\boldsymbol{T}_g) = \frac{\tau^2}{\tau^2 + \sigma^2} \boldsymbol{T}_g^{-1} \boldsymbol{A}^\top \boldsymbol{y}, \tag{25}$$

with $f(\boldsymbol{y}, \boldsymbol{A})$ being the special case where $\boldsymbol{T}_g = \boldsymbol{I}_n$. In the experiment, we set $n = 128 \times 128$ for a grayscale image with 128 rows and 128 columns and we compute the equivariance metric in eq. (24) (EQUIV) for the MAP reconstructor using 256 i.i.d. samples from the joint distribution. Table 9

shows the results for every angle and for the average over all angles. As predicted theoretically, perfect equivariance is achieved.

Table 9: **Empirical validation of the equivariance of MAP estimators.**

|  | 0° | 90° | 180° | 270° | Average |
|---|---|---|---|---|---|
| EQUIV | $\infty$ | $\infty$ | $\infty$ | $\infty$ | $\infty$ |

Table 10: **Training durations.** For each training, we report the average epoch duration and the number of epochs until the model is trained. Inpainting trainings are conducted on a single NVIDIA RTX 3090 Ti GPU and MRI trainings on a NVIDIA H100 GPU.

| | Image inpainting | |
|---|---|---|
| Method | Epoch duration (s) | Epochs |
| Supervised | 12 | 200 |
| ES (Ours) | 12 | 1000 |
| EI | 14 | 1000 |
| | MRI (×8 Accel., 40 dB SNR) | |
| Method | Epoch duration (s) | Epochs |
| Supervised | 29 | 200 |
| ES (Ours) | 24 | 13800 |
| EI | 53 | 7800 |
| SURE | 36 | 5700 |
| | MRI (×6 Accel., 10 dB SNR) | |
| Method | Epoch duration (s) | Epochs |
| Supervised | 19 | 70 |
| ES (Ours) | 19 | 3100 |
| EI | 75 | 2400 |
| SURE | 35 | 1200 |

## C    PROOFS

For the sake of clarity, we state the propositions and theorems a second time before their proofs. We also state and prove the additional Lemma 1 which helps prove Theorem 1.

**Lemma 1.** *The minimization problem*

$$\min_f \mathbb{E}_{\boldsymbol{x},\boldsymbol{y}} \left\{ \|\boldsymbol{A}f(\boldsymbol{y}) - \boldsymbol{A}\boldsymbol{x}\|^2 \right\} \tag{26}$$

*admits as solutions the functions of the form*

$$f(\boldsymbol{y}) = \boldsymbol{A}^{\dagger}\boldsymbol{A}\mathbb{E}_{\boldsymbol{x}|\boldsymbol{y}} \left\{ \boldsymbol{x} \right\} + (\boldsymbol{I} - \boldsymbol{A}^{\dagger}\boldsymbol{A})v(\boldsymbol{y}) \tag{27}$$

*where $v(\boldsymbol{y})$ is any function.*

*Proof.* Let's start by stating that $f(\boldsymbol{y}) = \mathbb{E}_{\boldsymbol{x}|\boldsymbol{y}} \left\{ \boldsymbol{x} \right\}$ is the only solution of (Klenke, 2008)

$$\min_f \mathbb{E}_{\boldsymbol{x},\boldsymbol{y}} \left\{ \|f(\boldsymbol{y}) - \boldsymbol{x}\|^2 \right\}. \tag{28}$$

For $f$ any solution of eq. (26), applying it to $\tilde{f}(\boldsymbol{y}) = \boldsymbol{A}f(\boldsymbol{y})$ and $\tilde{x} = \boldsymbol{A}\boldsymbol{x}$ gives

$$\boldsymbol{A}f(\boldsymbol{y}) = \boldsymbol{A}\mathbb{E}_{\boldsymbol{x}|\boldsymbol{y}} \left\{ \boldsymbol{x} \right\}, \tag{29}$$

and applying $f(\boldsymbol{y}) = \boldsymbol{A}^{\dagger}\boldsymbol{A}f(\boldsymbol{y}) + (\boldsymbol{I} - \boldsymbol{A}^{\dagger}\boldsymbol{A})f(\boldsymbol{y})$ with $v(\boldsymbol{y}) := f(\boldsymbol{y})$ yields eq. (27). Conversely, the objective in eq. (26) has the same value no matter the $f$ satisfying eq. (27) and since at least one of them is solution of eq. (26), they all are. $\square$

**Theorem 1.** *In the case of noiseless measurements with $p(\boldsymbol{x})$ $\mathcal{G}$-invariant (Assumption 1), if the matrix $\boldsymbol{Q_{A_1}} \triangleq \mathbb{E}_{g|\boldsymbol{A_1}} \{(\boldsymbol{AT}_g)^\top \boldsymbol{AT}_g\} \in \mathbb{R}^{n \times n}$ has full rank for some split $\boldsymbol{A_1}$, then the splitting method yields the same MMSE-optimal reconstructions as the supervised method, i.e.,*

$$f^*(\boldsymbol{y_1}, \boldsymbol{A_1}) = \mathbb{E}_{\boldsymbol{x}|\boldsymbol{y_1}, \boldsymbol{A_1}} \{\boldsymbol{x}\}. \tag{9}$$

*Proof.*

$$\mathbb{E}_{\boldsymbol{y}}\{\mathcal{L}_{\text{ES}}(\boldsymbol{y}, \boldsymbol{A}, f)\}$$
$$= \mathbb{E}_{\boldsymbol{y}} \left\{\mathbb{E}_g\{\mathbb{E}_{\boldsymbol{y_1}, \boldsymbol{A_1}|\boldsymbol{y}, \boldsymbol{AT}_g} \{\|\boldsymbol{AT}_g f(\boldsymbol{y_1}, \boldsymbol{A_1}) - \boldsymbol{y}\|^2\}\}\right\}$$
$$= \mathbb{E}_{\boldsymbol{y}, g}\{\mathbb{E}_{\boldsymbol{y_1}, \boldsymbol{A_1}|\boldsymbol{y}, g} \{\|\boldsymbol{AT}_g f(\boldsymbol{y_1}, \boldsymbol{A_1}) - \boldsymbol{y}\|^2\}\}$$
$$= \mathbb{E}_{\boldsymbol{y}, \boldsymbol{y_1}, \boldsymbol{A_1}, g}\{\|\boldsymbol{AT}_g f(\boldsymbol{y_1}, \boldsymbol{A_1}) - \boldsymbol{y}\|^2\}$$
$$= \mathbb{E}_{\boldsymbol{x}, \boldsymbol{y_1}, \boldsymbol{A_1}, g}\{\|\boldsymbol{AT}_g f(\boldsymbol{y_1}, \boldsymbol{A_1}) - \boldsymbol{AT}_g \boldsymbol{x}\|^2\}$$
$$= \mathbb{E}_{\boldsymbol{y_1}, \boldsymbol{A_1}, \boldsymbol{x}} \mathbb{E}_{g|\boldsymbol{y_1}, \boldsymbol{A_1}} \left\{\|\boldsymbol{AT}_g\big(f(\boldsymbol{y_1}, \boldsymbol{A_1}) - \boldsymbol{x}\big)\|^2\right\}$$
$$= \mathbb{E}_{\boldsymbol{y_1}, \boldsymbol{A_1}, \boldsymbol{x}}\{(f(\boldsymbol{y_1}, \boldsymbol{A_1}) - \boldsymbol{x})^\top \boldsymbol{Q_{A_1}}(f(\boldsymbol{y_1}, \boldsymbol{A_1}) - \boldsymbol{x})\},$$

where the third line use that $p(\boldsymbol{y_1}, \boldsymbol{A_1} \mid \boldsymbol{y}, \boldsymbol{AT}_g) = p(\boldsymbol{y_1}, \boldsymbol{A_1} \mid \boldsymbol{y}, g)$ as $\boldsymbol{A}$ is fixed. The fifth line use the noiseless measurements assumption and the invariance of the distribution $p(\boldsymbol{x})$. The last line uses definition of $\boldsymbol{Q_{A_1}}$. By applying Lemma 1, the global minimizer of the expected loss is given by:

$$f^*(\boldsymbol{y_1}, \boldsymbol{A_1}) = \boldsymbol{Q}_{\boldsymbol{A_1}}^\dagger \boldsymbol{Q_{A_1}} \mathbb{E}_{\boldsymbol{x}|\boldsymbol{y_1}, \boldsymbol{A_1}} \{\boldsymbol{x}\} + (\boldsymbol{I} - \boldsymbol{Q}_{\boldsymbol{A_1}}^\dagger \boldsymbol{Q_{A_1}})v(\boldsymbol{y_1}) \tag{30}$$

where $v : \mathbb{R}^n \to \mathbb{R}^n$ is any function. Moreover, since $\boldsymbol{Q}$ has full-rank, $\boldsymbol{Q}_{\boldsymbol{A_1}}^\dagger = \boldsymbol{Q}_{\boldsymbol{A_1}}^{-1}$ and then

$$f^*(\boldsymbol{y_1}, \boldsymbol{A_1}) = \mathbb{E}_{\boldsymbol{x}|\boldsymbol{y_1}, \boldsymbol{A_1}} \{\boldsymbol{x}\}. \tag{31}$$

$\square$

**Proposition 1.** *If the matrix $\bar{\boldsymbol{Q}}_{\boldsymbol{A}} \triangleq \mathbb{E}_{\boldsymbol{A_1}|\boldsymbol{A}} \{\boldsymbol{Q_{A_1}}\} \in \mathbb{R}^{n \times n}$ is invertible and $f$ minimizes $\mathbb{E}_{\boldsymbol{y}}\{\mathcal{L}_{ES}(\boldsymbol{y}, \boldsymbol{A}, f)\}$. Then the reconstruction function*

$$\overline{f}(\boldsymbol{y}, \boldsymbol{A}) \triangleq \mathbb{E}_{\boldsymbol{y_1}, \boldsymbol{A_1}|\boldsymbol{y}, \boldsymbol{A}} \left\{\bar{\boldsymbol{Q}}_{\boldsymbol{A}}^{-1} \boldsymbol{Q_{A_1}} f(\boldsymbol{y_1}, \boldsymbol{A_1})\right\} \tag{10}$$

*satisfies*

$$\overline{f}(\boldsymbol{y}, \boldsymbol{A}) = \mathbb{E}_{\boldsymbol{y_1}, \boldsymbol{A_1}|\boldsymbol{y}, \boldsymbol{A}} \left\{\bar{\boldsymbol{Q}}_{\boldsymbol{A}}^{-1} \boldsymbol{Q_{A_1}} \mathbb{E}_{\boldsymbol{x}|\boldsymbol{y_1}, \boldsymbol{A_1}} \{\boldsymbol{x}\}\right\}. \tag{11}$$

*where eq. (11) is a convex combination of MMSE estimators for different splittings.*

*Proof.* By applying eq. (30) to $f$ in the definition of $\overline{f}$ we obtain:

$$\overline{f}(\boldsymbol{y}, \boldsymbol{A}) = \mathbb{E}_{\boldsymbol{y_1}, \boldsymbol{A_1}|\boldsymbol{y}, \boldsymbol{A}} \left\{\bar{\boldsymbol{Q}}_{\boldsymbol{A}}^{-1} \boldsymbol{Q_{A_1}} \big(\boldsymbol{Q}_{\boldsymbol{A_1}}^\dagger \boldsymbol{Q_{A_1}} \mathbb{E}_{\boldsymbol{x}|\boldsymbol{y_1}, \boldsymbol{A_1}} \{\boldsymbol{x}\} + (\boldsymbol{I} - \boldsymbol{Q}_{\boldsymbol{A_1}}^\dagger \boldsymbol{Q_{A_1}})v(\boldsymbol{y_1})\big)\right\}$$

$$= \mathbb{E}_{\boldsymbol{y_1}, \boldsymbol{A_1}|\boldsymbol{y}, \boldsymbol{A}} \left\{\bar{\boldsymbol{Q}}_{\boldsymbol{A}}^{-1} \boldsymbol{Q_{A_1}} \mathbb{E}_{\boldsymbol{x}|\boldsymbol{y_1}, \boldsymbol{A_1}} \{\boldsymbol{x}\}\right\} + \mathbb{E}_{\boldsymbol{y_1}, \boldsymbol{A_1}|\boldsymbol{y}, \boldsymbol{A}} \left\{\bar{\boldsymbol{Q}}_{\boldsymbol{A}}^{-1} \boldsymbol{Q_{A_1}} \big((\boldsymbol{I} - \boldsymbol{Q}_{\boldsymbol{A_1}}^\dagger \boldsymbol{Q_{A_1}})v(\boldsymbol{y_1})\big)\right\}$$

$$= \mathbb{E}_{\boldsymbol{y_1}, \boldsymbol{A_1}|\boldsymbol{y}, \boldsymbol{A}} \left\{\bar{\boldsymbol{Q}}_{\boldsymbol{A}}^{-1} \boldsymbol{Q_{A_1}} \mathbb{E}_{\boldsymbol{x}|\boldsymbol{y_1}, \boldsymbol{A_1}} \{\boldsymbol{x}\}\right\}.$$

$\square$

**Corollary 1.** *In order for the matrices $\boldsymbol{Q_{A_1}}$ or $\bar{\boldsymbol{Q}}_{\boldsymbol{A}}$ to have full rank, it is necessary that $\boldsymbol{A}$ is not equivariant:*

$$\exists g \in \mathcal{G}, \boldsymbol{AT}_g \neq \boldsymbol{T}_g \boldsymbol{A}. \tag{13}$$

*Proof.* Let's assume by contradiction that $\boldsymbol{A}$ is equivariant with respect to $\boldsymbol{T}_g$

$$\boldsymbol{AT}_g = \boldsymbol{T}_g \boldsymbol{A}, \tag{32}$$

and let $\boldsymbol{x} \in \ker(\boldsymbol{A})$.

$$\boldsymbol{Q_{A_1}} \boldsymbol{x} = \left(\mathbb{E}_{g|\boldsymbol{A_1}} \{(\boldsymbol{AT}_g)^\top \boldsymbol{AT}_g\}\right)\boldsymbol{x}$$
$$= \mathbb{E}_{g|\boldsymbol{A_1}} \{(\boldsymbol{AT}_g)^\top \boldsymbol{AT}_g \boldsymbol{x}\}$$
$$= \mathbb{E}_{g|\boldsymbol{A_1}} \{(\boldsymbol{AT}_g)^\top \boldsymbol{T}_g \boldsymbol{A} \boldsymbol{x}\}$$
$$= \mathbb{E}_{g|\boldsymbol{A_1}} \{(\boldsymbol{AT}_g)^\top \boldsymbol{T}_g \boldsymbol{0}\}$$
$$= \boldsymbol{0}$$

Therefore,

$$\ker(\boldsymbol{Q}_{\boldsymbol{A}_1}) \supseteq \ker(\boldsymbol{A}) \supsetneq \{\boldsymbol{0}\}. \tag{33}$$

The matrix $\boldsymbol{Q}_{\boldsymbol{A}_1}$ has a non-trivial nullspace and thus cannot have full rank. Moreover, since this non-trivial nullspace is the same for all virtual operators $\boldsymbol{A}\boldsymbol{T}_g$, then $\bar{\boldsymbol{Q}}_{\boldsymbol{A}}$ shares the same non-trivial nullspace. $\qquad\square$

**Theorem 2.** *The reconstruction functions defined in points below are all equivariant as in eq. (15).*

1. **Artifact removal network.** *For a denoiser $\phi(\boldsymbol{x})$ equivariant in the sense of eq. (14),*

$$f(\boldsymbol{y}, \boldsymbol{A}) = \phi\left(\boldsymbol{A}^\top \boldsymbol{y}\right), \text{ or } f(\boldsymbol{y}, \boldsymbol{A}) = \phi\left(\boldsymbol{A}^\dagger \boldsymbol{y}\right). \tag{16}$$

2. **Unrolled network.** *For $\phi(\boldsymbol{x})$ equivariant, any $\gamma \in \mathbb{R}$ and data fidelity $d(\boldsymbol{A}\boldsymbol{x}, \boldsymbol{y})$, with*

$$\boldsymbol{x}_0 = \boldsymbol{0}, \quad \boldsymbol{x}_{k+1} = \phi\big(\boldsymbol{x}_k - \gamma\nabla_{\boldsymbol{x}_k}d(\boldsymbol{A}\boldsymbol{x}_k, \boldsymbol{y})\big) \tag{17}$$

*for $k = 0, \ldots, L-1$ and $f(\boldsymbol{y}, \boldsymbol{A}) = \boldsymbol{x}_L$.*

3. **Reynolds averaging.** *For a possibly non-equivariant reconstructor $r(\boldsymbol{y}, \boldsymbol{A})$, with*

$$f(\boldsymbol{y}, \boldsymbol{A}) = \frac{1}{|\mathcal{G}|} \sum_{g \in \mathcal{G}} \boldsymbol{T}_g r(\boldsymbol{y}, \boldsymbol{A}\boldsymbol{T}_g). \tag{18}$$

4. **Maximum a posteriori (MAP).** *For a distribution $p(\boldsymbol{x})$ invariant as in eq. (5), with*

$$f(\boldsymbol{y}, \boldsymbol{A}) = \underset{\boldsymbol{x} \in \mathbb{R}^n}{\operatorname{argmax}} \left\{ p(\boldsymbol{x} \mid \boldsymbol{y}, \boldsymbol{A}) \right\}. \tag{19}$$

5. **Minimum mean squared error (MMSE).** *For a distribution $p(\boldsymbol{x})$ invariant as in eq. (5),*

$$f(\boldsymbol{y}, \boldsymbol{A}) = \mathbb{E}_{\boldsymbol{x}|\boldsymbol{y}, \boldsymbol{A}} \left\{ \boldsymbol{x} \right\}. \tag{20}$$

*Proof.* We prove each case separately.

1. Denoting $\boldsymbol{A}^\times := \boldsymbol{A}^\top$ or $\boldsymbol{A}^\times := \boldsymbol{A}^\dagger$, eq. (16) gives, as $(\boldsymbol{A}\boldsymbol{T}_g)^\times = \boldsymbol{T}_g^{-1}\boldsymbol{A}^\times$,

$$f(\boldsymbol{y}, \boldsymbol{A}\boldsymbol{T}_g) = \phi\left(\boldsymbol{T}_g^{-1}\boldsymbol{A}^\times\boldsymbol{y}\right), \tag{34}$$

and since $\phi(\boldsymbol{x})$ is equivariant, i.e., eq. (14) holds, it simplifies to eq. (15).

2. We start by making the notation show the explicit dependency on $\boldsymbol{y}$ and $\boldsymbol{A}$:

$$\boldsymbol{x}_0(\boldsymbol{y}, \boldsymbol{A}) = \boldsymbol{0}, \quad \boldsymbol{x}_{k+1}(\boldsymbol{y}, \boldsymbol{A}) = \phi\Big(\boldsymbol{x}_k(\boldsymbol{y}, \boldsymbol{A}) - \gamma\nabla_{\boldsymbol{x}_k(\boldsymbol{y}, \boldsymbol{A})}d(\boldsymbol{A}\boldsymbol{x}_k(\boldsymbol{y}, \boldsymbol{A}), \boldsymbol{y})\Big). \tag{35}$$

and proceed to show that for $k = 0, \ldots, L$ it holds that

$$\boldsymbol{x}_k(\boldsymbol{y}, \boldsymbol{A}\boldsymbol{T}_g) = \boldsymbol{T}_g^{-1}\boldsymbol{x}_k(\boldsymbol{y}, \boldsymbol{A}). \tag{36}$$

For $k = 0$, it holds as

$$\boldsymbol{x}_0(\boldsymbol{y}, \boldsymbol{A}\boldsymbol{T}_g) = \boldsymbol{0} = \boldsymbol{T}_g^{-1}\boldsymbol{x}_0(\boldsymbol{y}, \boldsymbol{A}). \tag{37}$$

Let's assume that eq. (36) holds for $k < L$. Applying it and the chain rule in eq. (35) yields

$$\boldsymbol{x}_{k+1}(\boldsymbol{y}, \boldsymbol{A}\boldsymbol{T}_g) = \phi\Big(\boldsymbol{T}_g^{-1}\left(x_k(\boldsymbol{y}, \boldsymbol{A}) - \gamma\nabla_{\boldsymbol{x}_k(\boldsymbol{y}, \boldsymbol{A})}d(\boldsymbol{A}\boldsymbol{x}_k(\boldsymbol{y}, \boldsymbol{A}), \boldsymbol{y})\right)\Big). \tag{38}$$

Finally, applying eq. (14) in this equation gives

$$\boldsymbol{x}_{k+1}(\boldsymbol{y}, \boldsymbol{A}\boldsymbol{T}_g) = \boldsymbol{T}_g^{-1}\boldsymbol{x}_{k+1}(\boldsymbol{y}, \boldsymbol{A}), \tag{39}$$

and by induction, as $f(\boldsymbol{y}, \boldsymbol{A}) = \boldsymbol{x}_L(\boldsymbol{y}, \boldsymbol{A})$, eq. (15) holds.

3. From eq. (18), it holds that

$$f(\boldsymbol{y}, \boldsymbol{A}\boldsymbol{T}_g) = \frac{1}{|\mathcal{G}|} \sum_{h \in \mathcal{G}} \boldsymbol{T}_h r(\boldsymbol{y}, \boldsymbol{A}\boldsymbol{T}_g \boldsymbol{T}_h), \tag{40}$$

which, as the group action property holds $T_g T_h = T_{gh}$, rewrites as

$$f(\boldsymbol{y}, \boldsymbol{A}\boldsymbol{T}_g) = \frac{1}{|\mathcal{G}|} \sum_{h \in \mathcal{G}} \boldsymbol{T}_h r(\boldsymbol{y}, \boldsymbol{A}\boldsymbol{T}_{gh}), \tag{41}$$

Applying the change of variable $h' = gh$ in this equation gives

$$f(\boldsymbol{y}, \boldsymbol{A}\boldsymbol{T}_g) = \frac{1}{|\mathcal{G}|} \sum_{h \in \mathcal{G}} \boldsymbol{T}_{g^{-1}h} r(\boldsymbol{y}, \boldsymbol{A}\boldsymbol{T}_h), \tag{42}$$

which finally, using group action property again $\boldsymbol{T}_{g^{-1}h} = \boldsymbol{T}_g^{-1}\boldsymbol{T}_g$, gives eq. (15).

4. Taking the negative natural logarithm in eq. (19) and using Bayes' theorem gives

$$f(\boldsymbol{y}, \boldsymbol{A}) = \operatorname*{argmin}_{\boldsymbol{x} \in \mathbb{R}^n} \left\{ d(\boldsymbol{A}\boldsymbol{x}, \boldsymbol{y}) + \rho(\boldsymbol{x}) \right\}, \tag{43}$$

where $d(\boldsymbol{A}\boldsymbol{x}, \boldsymbol{y}) = -\log p(\boldsymbol{y} \mid \boldsymbol{A}\boldsymbol{x})$ and $\rho(\boldsymbol{x}) = -\log p(\boldsymbol{x})$. Applying $x' = \boldsymbol{T}_g\,\boldsymbol{x}$,

$$f(\boldsymbol{y}, \boldsymbol{A}\boldsymbol{T}_g) = \boldsymbol{T}_g^{-1} \operatorname*{argmin}_{\boldsymbol{x} \in \mathbb{R}^n} \left\{ d(\boldsymbol{A}\boldsymbol{x}, \boldsymbol{y}) + \rho\left(\boldsymbol{T}_g^{-1}\boldsymbol{x}\right) \right\}, \tag{44}$$

and eq. (5) makes $\rho(\boldsymbol{x})$ invariant as well $\rho\left(\boldsymbol{T}_g^{-1}\boldsymbol{x}\right) = \rho(\boldsymbol{x})$. Therefore, eq. (15) holds.

5. Let's assume that $p(\boldsymbol{x})$ and $p(\boldsymbol{A})$ are invariant in the sense of eq. (5). We first prove that

$$p(\boldsymbol{y}, \boldsymbol{A}\boldsymbol{T}_g) = p(\boldsymbol{y}, \boldsymbol{A}). \tag{45}$$

Using eq. (5), the invariance of $p(\boldsymbol{A})$ and the independence of $\boldsymbol{x}$ and $\boldsymbol{A}$, we compute

$$\begin{aligned} p(\boldsymbol{y}, \boldsymbol{A}\boldsymbol{T}_g) &= \mathbb{E}_{\boldsymbol{x}} \left\{ p(\boldsymbol{y}, \boldsymbol{A}\boldsymbol{T}_g \mid \boldsymbol{x}) \right\} = \mathbb{E}_{\boldsymbol{x}} \left\{ p(\boldsymbol{y}, \mid \boldsymbol{A}\boldsymbol{T}_g, \boldsymbol{x}) p(\boldsymbol{A}\boldsymbol{T}_g \mid \boldsymbol{x}) \right\} \\ &= \mathbb{E}_{\boldsymbol{x}} \left\{ p(\boldsymbol{y} \mid \boldsymbol{A}\boldsymbol{T}_g, \boldsymbol{x}) p(\boldsymbol{A}\boldsymbol{T}_g) \right\} = \mathbb{E}_{\boldsymbol{x}} \left\{ p(\boldsymbol{y} \mid \boldsymbol{A}\boldsymbol{T}_g\boldsymbol{x}) p(\boldsymbol{A}) \right\} \\ &= \mathbb{E}_{\boldsymbol{x}} \left\{ p(\boldsymbol{y} \mid \boldsymbol{A}\boldsymbol{x}) p(\boldsymbol{A}) \right\} = \mathbb{E}_{\boldsymbol{x}} \left\{ p(\boldsymbol{y} \mid \boldsymbol{A}, \boldsymbol{x}) p(\boldsymbol{A} \mid \boldsymbol{x}) \right\} \\ &= \mathbb{E}_{\boldsymbol{x}} \left\{ p(\boldsymbol{y}, \boldsymbol{A} \mid \boldsymbol{x}) \right\} = p(\boldsymbol{y}, \boldsymbol{A}). \end{aligned}$$

Next we start from

$$f(\boldsymbol{y}, \boldsymbol{A}\boldsymbol{T}_g) = \mathbb{E}_{\boldsymbol{x} \mid \boldsymbol{y}, \boldsymbol{A}\boldsymbol{T}_g} \left\{ \boldsymbol{x} \right\}. \tag{46}$$

and applying the integral formula for expectations gives

$$f(\boldsymbol{y}, \boldsymbol{A}\boldsymbol{T}_g) = \int \boldsymbol{x}\, p(\boldsymbol{x} \mid \boldsymbol{y}, \boldsymbol{A}\boldsymbol{T}_g)\, \mathrm{d}x. \tag{47}$$

Using Bayes' formula and $p(\boldsymbol{y} \mid \boldsymbol{A}, \boldsymbol{x}) = p(\boldsymbol{y} \mid \boldsymbol{A}\boldsymbol{x})$, it becomes

$$f(\boldsymbol{y}, \boldsymbol{A}\boldsymbol{T}_g) = \int \boldsymbol{x}\, \frac{p(\boldsymbol{y} \mid \boldsymbol{A}\boldsymbol{T}_g\boldsymbol{x})}{p(\boldsymbol{A}\boldsymbol{T}_g, \boldsymbol{y})}\, p(\boldsymbol{A}\boldsymbol{T}_g, \boldsymbol{x})\, \mathrm{d}x. \tag{48}$$

By using eq. (45), the invariance of $p(\boldsymbol{A})$ and the independence of $\boldsymbol{A}$ with $\boldsymbol{x}$, we obtain

$$f(\boldsymbol{y}, \boldsymbol{A}\boldsymbol{T}_g) = \int \boldsymbol{x}\, \frac{p(\boldsymbol{y} \mid \boldsymbol{A}\boldsymbol{T}_g\boldsymbol{x})}{p(\boldsymbol{A}, \boldsymbol{y})}\, p(\boldsymbol{A})p(\boldsymbol{x})\, \mathrm{d}x. \tag{49}$$

With the change of variable $\boldsymbol{x}' = \boldsymbol{T}_g\boldsymbol{x}$, and since $\boldsymbol{T}_g$ is unitary, we arrive at

$$f(\boldsymbol{y}, \boldsymbol{A}\boldsymbol{T}_g) = \int \boldsymbol{T}_g^{-1}\boldsymbol{x}\, \frac{p(\boldsymbol{y} \mid \boldsymbol{A}\boldsymbol{x})}{p(\boldsymbol{A}, \boldsymbol{y})}\, p(\boldsymbol{A})p(\boldsymbol{T}_g^{-1}\boldsymbol{x})\, \mathrm{d}x. \tag{50}$$

Finally, applying eq. (5) in this equation yields eq. (15).

$\square$

**Theorem 3.** *If $f(\boldsymbol{y}, \boldsymbol{A})$ is an equivariant reconstructor, then ES is equivalent to the splitting loss*

$$\mathcal{L}_{\text{ES}}(\boldsymbol{y}, \boldsymbol{A}, f) = \mathcal{L}_{\text{SPLIT}}(\boldsymbol{y}, \boldsymbol{A}, f). \tag{21}$$

*Proof.* We start from eq. (7)

$$\mathcal{L}_{\text{ES}}(\boldsymbol{y}, \boldsymbol{A}, f) \triangleq \mathbb{E}_g \left\{ \mathcal{L}_{\text{SPLIT}}(\boldsymbol{y}, \boldsymbol{A}\boldsymbol{T}_g, f) \right\} \tag{51}$$

$$= \mathbb{E}_g \left\{ \mathbb{E}_{\boldsymbol{y}_1, \boldsymbol{A}_1 | \boldsymbol{y}, \boldsymbol{A}\boldsymbol{T}_g} \left\{ \| \boldsymbol{A}\boldsymbol{T}_g f(\boldsymbol{y}_1, \boldsymbol{A}_1) - \boldsymbol{y} \|^2 \right\} \right\} \tag{52}$$

As $\boldsymbol{A}_1$ is a splitting of $\boldsymbol{A}\boldsymbol{T}_g$, we can write $\boldsymbol{A}_1 = \boldsymbol{M}\boldsymbol{A}\boldsymbol{T}_g$ for $\boldsymbol{M}$ a splitting matrix. We obtain

$$\mathcal{L}_{\text{ES}}(\boldsymbol{y}, \boldsymbol{A}, f) = \mathbb{E}_g \left\{ \mathbb{E}_{\boldsymbol{M} | \boldsymbol{y}, g} \left\{ \| \boldsymbol{A}\boldsymbol{T}_g f(\boldsymbol{M}\boldsymbol{y}, \boldsymbol{M}\boldsymbol{A}\boldsymbol{T}_g) - \boldsymbol{y} \|^2 \right\} \right\} \tag{53}$$

Applying eq. (15) and cancelling out $\boldsymbol{T}_g$ with $\boldsymbol{T}_g^{-1}$ yields

$$\mathcal{L}_{\text{ES}}(\boldsymbol{y}, \boldsymbol{A}, f) = \mathbb{E}_g \left\{ \mathbb{E}_{\boldsymbol{M} | \boldsymbol{y}, g} \left\{ \| \boldsymbol{A} f(\boldsymbol{M}\boldsymbol{y}, \boldsymbol{M}\boldsymbol{A}) - \boldsymbol{y} \|^2 \right\} \right\}. \tag{54}$$

By dropping the expectation in $g$ and rewriting $(\boldsymbol{M}\boldsymbol{y}, \boldsymbol{M}\boldsymbol{A})$ as $(\boldsymbol{y}_1, \boldsymbol{A}_1)$, this yields in eq. (21). $\quad\square$

## REPRODUCIBILITY STATEMENT

We share the implementation of our method and experiments (code in Section 1) to make our work easier to reproduce.

