# OpenReview forum: "Equivariant Splitting: Self-supervised learning from incomplete data"
_ICLR.cc/2026/Conference — ICLR 2026 Poster_

### Official Review · Reviewer_vQFi · 2025-10-30

**Soundness:** 2
**Presentation:** 2
**Contribution:** 2
**Rating:** 4
**Confidence:** 2

**Summary:**

The paper proposes Equivariant Splitting (ES), a self-supervised training loss for inverse problems that combines measurement-splitting ideas with equivariant modelling. Under an invariance assumption on the image distribution (Assumption 1), ES averages splitting losses computed on virtual operators formed by composing the forward operator with group transforms and shows (Theorem 1 / Proposition 1) that, when certain matrix conditions hold, minimizers of the ES loss coincide (in expectation) with the MMSE estimator. The authors introduce a reconstructor notion of equivariance (Definition 1), prove that several common architecture families satisfy it (Theorem 2), and show ES reduces to standard splitting for equivariant reconstructors (Theorem 3). Empirical evaluation covers compressive sensing, image inpainting, and MRI, comparing ES to supervised, equivariant imaging (EI), SURE baselines, and reporting PSNR / SSIM / an equivariance metric.

**Strengths:**

- The paper formulates a unifying view that connects splitting losses and equivariant imaging and states the assumptions under which ES is theoretically unbiased for the supervised objective.
- Introducing a definition of equivariant reconstructors and enumerating common architectures that satisfy it is useful.
- The proposed loss is practically implementable. The paper explains how to use Monte-Carlo sampling of transforms and splits, and how to replace the noiseless term with R2R in the noisy setting, making the method applicable to realistic measurement noise models.
- Experiments span multiple inverse problems (compressive sensing, inpainting, MRI) and include an ablation on equivariant vs non-equivariant architectures, which demonstrates the claimed synergy between splitting losses and equivariant architectures in the tested settings.

**Weaknesses:**

- The main theoretical guarantee requires (i) the distributional invariance (Assumption 1) and (ii) invertibility conditions (full rank) on the matrix $Q_{A_1}$ or $\bar{Q}_A$ for some split $A_1$. These conditions are sufficient but seem strong and may rarely hold in practice; the manuscript gives limited practical diagnostics or guidance on verifying these conditions for real forward operators.
- The method depends critically on choosing an appropriate group (translations, rotations, flips, etc.). The paper does not sufficiently analyze robustness when the invariance assumption is only approximate or mis-specified (e.g., natural images that are not strictly invariant to some transforms). Practical recommendations for selecting $G$ per application are brief.
- Although multiple tasks are included, some experimental choices raise concerns: (i) use of MNIST for compressive sensing (very simple data distribution), (ii) DIV2K for inpainting with synthetic masks, and (iii) synthetically generated k-space masks for MRI. The assertion that ES achieves “state-of-the-art” self-supervised performance is not fully established across diverse, realistic datasets or against the latest baselines in unsupervised/self-supervised imaging literature (such as DDRM).
- Reynolds averaging over large groups is noted to be impractical, and the authors use Monte-Carlo sampling, but the runtime, per-iteration cost, and the number of samples are not clearly reported. Training budgets are mentioned (up to 50 hours on a single GPU), but fair comparisons to baselines in terms of wall-clock cost or memory are missing.
- Ablations and failure modes are not discussed in detail. The ablation on equivariant architectures is informative, but there is little analysis of when ES fails (e.g., non-unitary transforms or operators that are nearly equivariant. Corollary 1 suggests A must not commute with transforms).

**Questions:**

- In practice, how does one verify (or have you measured) whether $Q_{A_1}$ or $\bar{Q}_A$ is full rank for realistic forward operators (e.g., the MRI coil-sensitivity map that you use)?
- How sensitive is ES when the dataset only approximately satisfies Assumption 1? Can you provide quantitative experiments where invariance is gradually violated (e.g., introduce systematic asymmetries) and show performance degradation versus EI / supervised baselines?
- What is the typical number of random transforms/splits used at training and at test time? What is the overhead relative to EI and supervised training?
- Corollary 1 requires that the forward operator $A$ is not equivariant. Can you clarify and demonstrate what happens when $A$ is equivariant or nearly equivariant (e.g., radial sampling in MRI, which is equivariant with rotational transforms)? Does ES break down or merely degrade gracefully?
- Reproducibility and released artifacts. The paper references DeepInverse and gives some training details, but can you (i) release code and trained models, (ii) provide exact seeds and the split/sampling scripts for splits/transforms, and (iii) include scripts to reproduce the key tables (Tables 1–4) and the equivariance metric computations (EQUIV)? Appendix B mentions some implementation choices, but the reproducibility checklist is incomplete.

---

> ### Author Response · Authors · 2025-11-21
> **Response part 1**
>
> Thank you for your comments and suggestions. We address them below and in the revised manuscript with the changes in red.
>
> > The method depends critically on choosing an appropriate group (translations, rotations, flips, etc.). The paper does not sufficiently analyze robustness when the invariance assumption is only approximate or mis-specified (e.g., natural images that are not strictly invariant to some transforms).
>
> > How sensitive is ES when the dataset only approximately satisfies Assumption 1? Can you provide quantitative experiments where invariance is gradually violated (e.g., introduce systematic asymmetries) and show performance degradation versus EI / supervised baselines?
>
> Although our method is theoretically motivated by invariance assumptions, our experiments show that the proposed loss performs well even when these assumptions hold only approximately.  Notably, our experiments use **real-world image datasets that are not perfectly invariant** (nor they have been made invariant in a synthetic way, as often done in equivariance-related papers). Most digits in MNIST and objects in DIV2K scenes are located at the center of the image, and most knee scans in FastMRI share a similar vertical orientation. Despite the datasets not being exactly invariant, our experiments show that the proposed loss remains effective at learning from data associated with a single rank-deficient operator.
>
>
> > Practical recommendations for selecting $G$ per application are brief.
>
> We have added in Appendix A.2 of the updated manuscript, a paragraph to help determine how to select the group action. It can be chosen based on _(i)_ the natural invariances of the data and (ii) whether the forward operator $A$ is equivariant to a given transformation—following Corollary 1, one should favor transformations for which $A$ is not equivariant. Table 5 of the updated manuscript (see a copy below) provides a practical summary for common operators.
>
> | Operator            | Translation | Rotation | Permutation | Amplitude |
> | ------------------- | ----------- | -------- | ----------- | --------- |
> | Isotropic blur      | ✗           | ✗        | ✓           | ✗         |
> | Image inpainting    | ✓           | ✓        | ✓           | ✗         |
> | Sparse-view CT      | ✗           | ✓        | ✓           | ✗         |
> | Accelerated MRI     | ✗           | ✓        | ✓           | ✗         |
> | Compressive sensing | ✓           | ✓        | ✓           | ✗         |
>
> **Table 5: Decision table for the transformations.** Corollary 1 shows that not all transformations are well-suited for all problems. Namely, transformations for which the operator is equivariant introduce no additional information for learning in the nullspace of the operator. This table specifies which transformations are well-suited for which operator.
>
> > The main theoretical guarantee requires (i) the distributional invariance (Assumption 1) and (ii) invertibility conditions (full rank) on the matrix  $Q_{A_1}$ or $\bar{Q}_{A_1}$ for some split $A_1$. These conditions are sufficient but seem strong and may rarely hold in practice; the manuscript gives limited practical diagnostics or guidance on verifying these conditions for real forward operators.

---

> ### Author Response · Authors · 2025-11-21
> **Response part 2**
>
> > Practical recommendations for selecting $G$ per application are brief.
>
> We have added in Appendix A.2 of the updated manuscript, a paragraph to help determine how to select the group action. It can be chosen based on _(i)_ the natural invariances of the data and (ii) whether the forward operator $A$ is equivariant to a given transformation—following Corollary 1, one should favor transformations for which $A$ is not equivariant. Table 5 of the updated manuscript (see a copy below) provides a practical summary for common operators.
>
> | Operator            | Translation | Rotation | Permutation | Amplitude |
> | ------------------- | ----------- | -------- | ----------- | --------- |
> | Isotropic blur      | ✗           | ✗        | ✓           | ✗         |
> | Image inpainting    | ✓           | ✓        | ✓           | ✗         |
> | Sparse-view CT      | ✗           | ✓        | ✓           | ✗         |
> | Accelerated MRI     | ✗           | ✓        | ✓           | ✗         |
> | Compressive sensing | ✓           | ✓        | ✓           | ✗         |
>
> **Table 5: Decision table for the transformations.** Corollary 1 shows that not all transformations are well-suited for all problems. Namely, transformations for which the operator is equivariant introduce no additional information for learning in the nullspace of the operator. This table specifies which transformations are well-suited for which operator.
>
>
> > The main theoretical guarantee requires (i) the distributional invariance (Assumption 1) and (ii) invertibility conditions (full rank) on the matrix  $Q_{A_1}$ or $\bar{Q}_{A_1}$ for some split $A_1$. These conditions are sufficient but seem strong and may rarely hold in practice; the manuscript gives limited practical diagnostics or guidance on verifying these conditions for real forward operators.
>
> > In practice, how does one verify (or have you measured) whether $Q_{A_1}$ or  $\bar{Q}_{A_1}$ is full rank for realistic forward operators (e.g., the MRI coil-sensitivity map that you use)?
>
> We would like to emphasise that the difficulty of computing $Q_{A_1}$ is not a practical limitation. It is mainly a theoretical condition, while in practice empirical validation prevails: as explained in the answers above, we obtain good empirical performance even if the real dataset is not exactly invariant, and we provide a set of guidelines to choose a transform that is well-suited for various inverse problems.
>
> Moreover, while the full-rank assumption provides a rigorous mathematical guarantee, in practice, the performance will depend on the spectrum of $Q_{A_1}$ or $\bar Q_{A_1}$: matrices that have very small eigenvalues (thus being close to singular) will perform worse than matrices with a flatter spectrum. We have included this discussion in the updated manuscript (after the statement of Theorem 1).
>
> In practice, one could compute (approximations of) the spectrum of $Q_{A_1}$ or $\bar Q_{A_1}$ in various ways:
>
> i) If the matrix can be explicitly instantiated (in the case of problems with small images and operators), one can directly compute its rank via an SVD decomposition.
>
> ii) When the matrix cannot be explicitly formed due to its large-scale nature, Algorithm 4.4 in (Halko et al., 2009) provides a practical matrix-free way (in the sense that one does not need to perform matrix-vector multiplication to evaluate $Q_{A_1}x$) to compute a matrix $Q$ whose range approximates that of $\bar Q_{A_1}$. In our situation, we could approximate $Q_{A_1}x$ by an empirical expectation $$Q_{A_1}x \approx \frac{1}{|\tilde{G}|}\sum_{g\in \tilde{G}} (AT_g)^\top AT_g p(g | A_1) x,$$ where $\tilde{G}$ is a random subset of the whole group $G$, still in a matrix-free manner.
>
>
> - Halko, N., Martinsson, P. G., & Tropp, J. A. (2011). Finding structure with randomness: Probabilistic algorithms for constructing approximate matrix decompositions. SIAM review, 53(2), 217-288.

---

> ### Author Response · Authors · 2025-11-21
> **Response part 3**
>
> >Although multiple tasks are included, some experimental choices raise concerns: (i) use of MNIST for compressive sensing (very simple data distribution), (ii) DIV2K for inpainting with synthetic masks, and (iii) synthetically generated k-space masks for MRI. The assertion that ES achieves “state-of-the-art” self-supervised performance is not fully established across diverse, realistic datasets or against the latest baselines in unsupervised/self-supervised imaging literature (such as DDRM).
>
> We have included an additional experiment in the appendix, reported in Table 7 copied below, that covers more realistic inpainting masks proposed by the reviewer. The setup corresponds to a noisy inpainting problem where the mask randomly removes entire columns, which is representative of certain satellite imaging conditions, e.g., push-broom scanners (Xu et al. 2016).  As reported in Table 7, the performance remains comparable to that of the EI baseline.
>
> DDRM and most diffusion methods are not self-supervised, as they rely on a pre-trained denoiser that has been trained on ground-truth references. For example, the DDRM paper relies on a denoiser working on color images and cannot be deployed on complex images without re-training another denoiser (e.g., for our MRI experiments).  Nonetheless, recently some works have shown that it is possible to train a diffusion model without ground-truth data (Daras et al, 2023). These works typically rely on splitting losses using data associated to multiple forward operators (such as varying masks across MRI scans) to train the model. Since our work extends splitting losses to the case of a *single* forward operator, we believe that it will pave the way for training diffusion models in the more realistic setting where data is observed via a single operator.
>
>
> | Push-broom mask                                           | Method           |    PSNR ↑    |     SSIM ↑    |
> | :------------------------------------------------- | :--------------- | :----------: | :-----------: |
> | Randomly-spaced | Supervised       | 23.72 ± 2.10 | 0.743 ± 0.051 |
> |                                                    | ES (Ours)        | 23.04 ± 1.81 | 0.734 ± 0.050 |
> |                                                    | EI               | 23.05 ± 2.13 | 0.707 ± 0.066 |
> |                                                    | Incomplete image |  9.44 ± 2.31 | 0.141 ± 0.048 |
> | Evenly-spaced   | Supervised       | 28.37 ± 2.15 | 0.873 ± 0.035 |
> |                                                    | ES (Ours)        | 21.94 ± 2.12 | 0.617 ± 0.091 |
> |                                                    | Incomplete image |  9.61 ± 2.42 | 0.152 ± 0.061 |
>
> **Table 7. Inpainting with push-broom masks.** We consider an unevenly-spaced mask sampled randomly to test our method in a realistic inpainting setting, and an evenly-spaced mask to verify the claim in Corollary 1 empirically in a case of almost-equivariance.
>
> - Xu, X., Zhang, H., Han, G., Kwan, K. C., Pang, W. M., Fang, J., & Zhao, G. (2016). A Two-Phase Space Resection Model for Accurate Topographic Reconstruction from Lunar Imagery with PushbroomScanners. Sensors, 16(4), 507.
> - Daras, Giannis, et al. "Ambient diffusion: Learning clean distributions from corrupted data." Advances in Neural Information Processing Systems 36 (2023): 288-313.

---

> ### Author Response · Authors · 2025-11-21
> **Response part 4**
>
> > Reynolds averaging over large groups is noted to be impractical, and the authors use Monte-Carlo sampling, but the runtime, per-iteration cost, and the number of samples are not clearly reported. Training budgets are mentioned (up to 50 hours on a single GPU), but fair comparisons to baselines in terms of wall-clock cost or memory are missing.
>
> > What is the typical number of random transforms/splits used at training and at test time? What is the overhead relative to EI and supervised training?
>
> For the inpainting and compressive sensing experiments, we used 1 split per image per batch for training and 10 per image for testing. No averaging over transformations was done since we used an equivariant reconstructor.
>
> For the MRI experiments, we used 1 sample of $A_1$ (corresponding to 1 split and 1 transformation) per image per batch during training, and 10 during testing. This information has been added to the manuscript (page 5).
>
> Our method does not incur a training overhead when compared to EI. In fact, the opposite is true, as EI requires two to three evaluations of the network, involving higher memory and time costs. Compared to the supervised approach, the inpainting experiment shows nearly identical epoch times, with only a slight increase of 0.06 seconds per epoch. However, our method typically requires more training iterations to converge— as is common for most self-supervised methods—since there is a reduced amount of available information in the dataset. The results are shown in Table 11, see below for a copy of it.
>
>
> | Modality                   | Methods    | Epoch duration (s) | Epochs |
> |:-------------------------- |:---------- |:------------------:|:------:|
> | Image inpainting           | Supervised |         12         |  200   |
> |                            | ES (Ours)  |         12         |  1000  |
> |                            | EI         |         14         |  1000  |
> | MRI (x8 Accel., 40 dB SNR) | Supervised |         29         |  200   |
> |                            | ES (Ours)  |         24         | 13800  |
> |                            | EI         |         53         |  7800  |
> |                            | SURE       |         36         |  5700  |
> | MRI (x6 Accel., 10 dB SNR) | Supervised |         19         |   70   |
> |                            | ES (Ours)  |         19         |  3100  |
> |                            | EI         |         75         |  2400  |
> |                            | SURE       |         35         |  1200  |
>
> **Table 11. Training times.** For each training, we report the average epoch duration and the number of epochs until the model is trained. Inpainting trainings are conducted on a single NVIDIA RTX 3090 Ti GPU, and MRI trainings on an NVIDIA H100 GPU.

---

> ### Author Response · Authors · 2025-11-21
> **Response part 5**
>
> > Ablations and failure modes are not discussed in detail. The ablation on equivariant architectures is informative, but there is little analysis of when ES fails (e.g., non-unitary transforms or operators that are nearly equivariant. Corollary 1 suggests $A$ must not commute with transforms).
>
> > Corollary 1 requires that the forward operator $A$ is not equivariant. Can you clarify and demonstrate what happens when $A$ is equivariant or nearly equivariant (e.g., radial sampling in MRI, which is equivariant with rotational transforms)? Does ES break down or merely degrade gracefully?
>
> We added two additional inpainting experiments addressing your point:
> _(i)_ **Uniform push-broom scanners (PBS):** an experiment in which every other column of images are removed. In this case, the operator $A$ becomes _nearly_ equivariant in the sense that for any even horizontal shift $g$ we have $A T_g = T_gA$. This relation also holds for all vertical shifts. The results show a significant degradation in performance.
> _(ii)_ **Non-uniform PBS:** the experience is described in more detail in a previous point.
>
> The results show that performance (compared to the supervised baseline) drops significantly when the operator is very close to being equivariant (uniform PBS), but still performs well in cases with only approximate equivariance (random PBS). The results are included in Table 7, and we report part of them below.
>
> | Push-broom mask                                           | Method           |    PSNR ↑    |     SSIM ↑    |
> | :------------------------------------------------- | :--------------- | :----------: | :-----------: |
> | Randomly-spaced | Supervised       | 23.72 ± 2.10 | 0.743 ± 0.051 |
> |                                                    | ES (Ours)        | 23.04 ± 1.81 | 0.734 ± 0.050 |
> |                                                    | EI               | 23.05 ± 2.13 | 0.707 ± 0.066 |
> |                                                    | Incomplete image |  9.44 ± 2.31 | 0.141 ± 0.048 |
> | Evenly-spaced   | Supervised       | 28.37 ± 2.15 | 0.873 ± 0.035 |
> |                                                    | ES (Ours)        | 21.94 ± 2.12 | 0.617 ± 0.091 |
> |                                                    | Incomplete image |  9.61 ± 2.42 | 0.152 ± 0.061 |
>
> **Table 7. Inpainting with push-broom masks.** We consider a randomly-spaced mask sampled randomly to test our method in a realistic inpainting setting, and an evenly-spaced mask to verify the claim in Corollary 1 empirically in a case of almost-equivariance.
>
> >Reproducibility and released artifacts. The paper references DeepInverse and gives some training details, but can you (i) release code and trained models, (ii) provide exact seeds and the split/sampling scripts for splits/transforms, and (iii) include scripts to reproduce the key tables (Tables 1–4) and the equivariance metric computations (EQUIV)? Appendix B mentions some implementation choices, but the reproducibility checklist is incomplete.
>
> We agree that reproducible research is important and have made available the implementation of our method and experiments in a public, anonymous repository¹. We have also added the following reproducibility statement to the manuscript:
>
> _Reproducibility statement_
>
> _We share the implementation of our method and experiments¹ to make our work easier to reproduce._
>
> ¹ https://anonymous.4open.science/r/Equivariant-Splitting-ICLR2026/README.md
>
> Edit: there was a minor typo that we fixed.

---

### Official Review · Reviewer_ZYHS · 2025-10-31

**Soundness:** 2
**Presentation:** 2
**Contribution:** 2
**Rating:** 4
**Confidence:** 3

**Summary:**

This paper proposes Equivariant Splitting (ES), a self-supervised learning method for solving linear inverse problems when only incomplete measurements from a single forward operator are available. The method combines ideas from equivariant imaging (EI), which leverages invariance of the signal distribution under a group of transformations, and measurement splitting, which divides measurements into input/target pairs. The authors introduce a new definition of equivariance for reconstruction networks and show that, under mild conditions, minimizing their proposed loss yields the MMSE estimator. They validate ES on image inpainting, compressive sensing, and accelerated MRI, reporting performance close to supervised baselines and better than existing self-supervised methods like EI.

**Strengths:**

* The method targets a highly relevant and challenging problem: learning from incomplete data with a fixed forward operator (e.g., fixed MRI sampling mask, fixed inpainting mask).
* The reported results show consistent improvements over EI (especially in highly ill-posed regimes like high CS compression) and a performance closely following supervised learning methods.
* The ablation study in Table 3  demonstrates clearly the synergy between equivariant architectures and the ES loss, validating the theoretical claims.

**Weaknesses:**

* The proposed self-supervised learning strategy relies on strong assumptions on signal invariance and operator non-equivariance, while it does not explore what happens when these assumptions are violated.
* Limited Robustnes to real-world distortions: All experiments assume perferct knowledge of the forward operator A (in the linear inverse imaging literature this assumption is known as an "inverse crime”) and either idealized noise models (i.i.d Gaussian) or  the absence of noise. Morevoer, the studied problems are oversimplified and are not representative of problems met in real-world applications. For example, in image inpainting the authors consider a binary mask that keeps 30% of all the pixel and the absence of noise while more realistic scenarios would involve added noise and non-uniform masking.
* Computational Overhead and Scalability Concerns
  *  Although Theorem 3 shows that equivariant architectures reduce the ES loss to a standard splitting loss at training time, training still requires multiple random splits per sample, which increases memory and compute costs compared to simple supervised or consistency-based baselines.
  * The use of Reynolds averaging in Eq. 18 for enforcing equivariance becomes prohibitively expensive for large groups (e.g., continuous shifts or rotations), as noted in Appendix A. The paper only evaluates small discrete groups (e.g., 90° rotations + flips), limiting generalizability.

**Questions:**

Corollary 1 states that the forward operator A must not be equivariant w.r.t. $G$ for $Q_{A_1}$  to be full rank. In practice, how should one select $G$ given a fixed $A$ ? Are there automated or data-driven strategies to choose transformations that maximize the rank of $Q_{A_1}$?

---

> ### Author Response · Authors · 2025-11-21
> **Response part 1**
>
> Thank you for your comments and suggestions. We address them below and in the revised manuscript with the changes in red.
>
> > The proposed self-supervised learning strategy relies on strong assumptions on signal invariance and operator non-equivariance, while it does not explore what happens when these assumptions are violated.
>
> Although our method is theoretically motivated by invariance assumptions, our experiments show that the proposed loss performs well even when these assumptions hold only approximately.  Notably, our experiments use **real-world image datasets that are not perfectly invariant** (nor they have been made invariant in a synthetic way, as often done in equivariance-related papers). Most digits in MNIST and objects in DIV2K scenes are located at the center of the image, and most knee scans in FastMRI share a similar vertical orientation. Despite the datasets not being exactly invariant, our experiments show that the proposed loss remains effective at learning from data associated with a single rank-deficient operator.
>
> We agree nonetheless with the reviewer that we have not explored the case of nearly equivariant operators $A$. We have therefore added two inpainting experiments where the mask is chosen to be approximately equivariant to pixel shifts. See Appendix B.4 of the updated manuscript and Table 7 (copied below).
>
> - i) First, we assume that some randomly chosen columns of the image are not observed. In this setting, the operator $A$ is equivariant to vertical shifts: for any vertical shift indexed by $g$ we have that $AT_g =T_g A$ (and thus the nullspace of $A$ is not affected by the composition with $T_g$), but the operator is not equivariant to horizontal shifts. In this case, we observe that the proposed loss still achieves a good performance.
>
>
> - ii) Secondly, we assume that all the odd columns of the image are not observed. In this setting, the operator $A$ is equivariant to vertical shifts and almost equivariant to horizontal shifts: for any even horizontal shift indexed by $g$ we have that $AT_g = T_gA$ (and thus the nullspace of $A$ is not affected by the composition with $T_g$). Here, the results show poor performance.
>
> We hope this clarifies how the method behaves when the assumptions are not met.
>
> | Push-broom mask                                           | Method           |    PSNR ↑    |     SSIM ↑    |
> | :------------------------------------------------- | :--------------- | :----------: | :-----------: |
> | Randomly-spaced | Supervised       | 23.72 ± 2.10 | 0.743 ± 0.051 |
> |                                                    | ES (Ours)        | 23.04 ± 1.81 | 0.734 ± 0.050 |
> |                                                    | EI               | 23.05 ± 2.13 | 0.707 ± 0.066 |
> |                                                    | Incomplete image |  9.44 ± 2.31 | 0.141 ± 0.048 |
> | Evenly-spaced   | Supervised       | 28.37 ± 2.15 | 0.873 ± 0.035 |
> |                                                    | ES (Ours)        | 21.94 ± 2.12 | 0.617 ± 0.091 |
> |                                                    | Incomplete image |  9.61 ± 2.42 | 0.152 ± 0.061 |
>
> **Table 7. Inpainting with push-broom masks.** We consider an unevenly-spaced mask sampled randomly to test our method in a realistic inpainting setting, and an evenly-spaced mask to verify the claim in Corollary 1 empirically in a case of almost-equivariance.
>
> Edit: there was a minor typo that we fixed.

---

> ### Author Response · Authors · 2025-11-21
> **Response part 2**
>
> > Limited Robustnes to real-world distortions: All experiments assume perferct knowledge of the forward operator A (in the linear inverse imaging literature this assumption is known as an "inverse crime”) and either idealized noise models (i.i.d Gaussian) or the absence of noise.
>
> We agree that assuming complete knowledge of the forward operator $A$ can be restrictive in certain settings, e.g., in blind deblurring (Chakrabarti, 2016). Nevertheless, we believe that solving a blind inverse problem in a fully self-supervised manner remains a challenging task that is not yet solved in the literature. We also acknowledge that assuming a known noise distribution is a strong hypothesis that can potentially be relaxed. One possible direction, in cases where the noise family is known but its standard deviation is not (e.g., Gaussian noise with unknown standard deviation), is to replace the second term (R2R) in the proposed loss $\mathcal{L}_{G-ES}$ (page 5)
>
> $$\mathcal L_{\textrm{G-ES}} (y, A, f)  = \mathbb{E}_{g, y_1, A_1, \boldsymbol{\omega} \mid y, A T_g} \left\\{\left\\| A_1 f(y_1 + \alpha \boldsymbol{\omega}, A_1) - \Big(y_1 - \frac{\boldsymbol{\omega}}{\alpha}\Big) \right\\|^2  +  \\| A_2 f(y_1 + \alpha \boldsymbol{\omega}, A_1) - y_2\\|^2  \right\\}$$
> with a self-supervised denoising loss such as UNSURE (Tachella et al. 2024). When the noise model itself is unknown, one can instead test the robustness of the method with the ES loss (eq 7).
> We have therefore added the ablation study in Appendix B.4.1. The results displayed in Table 8 (copied below) in the updated manuscript show a degradation of approximately 1 dB for both the ES and EI methods, indicating that the method remains reasonably robust but does incur a performance drop under these relaxed assumptions.
>
> - Chakrabarti, A., 2016, September. A neural approach to blind motion deblurring. In European conference on computer vision (pp. 221-235). Cham: Springer International Publishing.
> - Tachella, J., Davies, M., & Jacques, L. (2024). UNSURE: self-supervised learning with Unknown Noise level and Stein's Unbiased Risk Estimate. arXiv preprint arXiv:2409.01985.
>
> | Modality                   | Method           | Known noise |    PSNR ↑    |      SSIM ↑     |
> | :------------------------- | :--------------- | :---: | :----------: | :-------------: |
> | Image inpainting           | Supervised       |       | 23.72 |  0.743  |
> |                            | ES (Ours)        |   ✓   | 23.04  |  0.734  |
> |                            | ES               |   ×   | 22.05  |   0.59 |
> |                            | EI               |   ✓   | 23.05  |  0.707 |
> |                            | EI               |   ×   | 22.01  |   0.59 |
> |                            | Incomplete image |       |  9.44  |  0.141 |
> | MRI (x8 Accel., 40 dB SNR) | Supervised       |       | 28.74  | 0.6445 |
> |                            | ES (Ours)        |   ✓   | 28.54  | 0.6195 |
> |                            | ES               |   ×   | 28.52  | 0.6194 |
> |                            | EI               |   ✓   | 27.88  | 0.5731 |
> |                            | EI               |   ×   | 27.89  | 0.5755 |
> |                            | IDFT             |       | 23.62  | 0.5052 |
> | MRI (x6 Accel., 10 dB SNR) | Supervised       |       | 27.39  | 0.5243 |
> |                            | ES (Ours)        |   ✓   | 27.33  | 0.5126 |
> |                            | ES               |   ×   | 25.73  | 0.4566 |
> |                            | EI               |   ✓   | 27.23  | 0.5110 |
> |                            | EI               |   ×   | 26.02  | 0.4706 |
> |                            | IDFT             |       | 23.85  | 0.3878 |
>
> **Table 8. Performance when the noise distribution is unknown.** In the unknown noise scenarios, we use the variants of ES and EI corresponding to assuming that the measurements are noiseless.

---

> ### Author Response · Authors · 2025-11-21
> **Response part 3**
>
> > Morevoer, the studied problems are oversimplified and are not representative of problems met in real-world applications. For example, in image inpainting the authors consider a binary mask that keeps 30% of all the pixel and the absence of noise while more realistic scenarios would involve added noise and non-uniform masking.
>
> We added in the appendix a more realistic experiment to address your point. It corresponds to noisy inpainting where entire columns are randomly removed, a scenario that arises in satellite imaging, e.g., push-broom scanners (Xu et al. 2016). Table 7 reports the results and shows that the performance is comparable to that of the EI baseline (see also a copy below).
>
> | Push-broom mask                                           | Method           |    PSNR ↑    |     SSIM ↑    |
> | :------------------------------------------------- | :--------------- | :----------: | :-----------: |
> | Randomly-spaced | Supervised       | 23.72 ± 2.10 | 0.743 ± 0.051 |
> |                                                    | ES (Ours)        | 23.04 ± 1.81 | 0.734 ± 0.050 |
> |                                                    | EI               | 23.05 ± 2.13 | 0.707 ± 0.066 |
> |                                                    | Incomplete image |  9.44 ± 2.31 | 0.141 ± 0.048 |
> | Evenly-spaced   | Supervised       | 28.37 ± 2.15 | 0.873 ± 0.035 |
> |                                                    | ES (Ours)        | 21.94 ± 2.12 | 0.617 ± 0.091 |
> |                                                    | Incomplete image |  9.61 ± 2.42 | 0.152 ± 0.061 |
>
> **Table 7. Inpainting with push-broom masks.** We consider an unevenly-spaced mask sampled randomly to test our method in a realistic inpainting setting, and an evenly-spaced mask to verify the claim in Corollary 1 empirically in a case of almost-equivariance.
>
> - Xu, X., Zhang, H., Han, G., Kwan, K. C., Pang, W. M., Fang, J., & Zhao, G. (2016). A Two-Phase Space Resection Model for Accurate Topographic Reconstruction from Lunar Imagery with Pushbroom Scanners. Sensors, 16(4), 507.
>
>
>
> > Computational Overhead and Scalability Concerns
>     - Although Theorem 3 shows that equivariant architectures reduce the ES loss to a standard splitting loss at training time, training still requires multiple random splits per sample, which increases memory and compute costs compared to simple supervised or consistency-based baselines.
>
> For the splitting, we use Monte Carlo sampling during training, which does not increase the memory cost. This contrasts with the EI baseline, which requires a double evaluation of the network and therefore leads to higher memory usage. We now report the corresponding values for all methods in the inpainting and MRI scenarios.
> We observe that ES does not take significantly more time per epoch than the supervised method (it only requires one additional application of the operator, which is fast compared to the network evaluation). However, it typically requires more training iterations— as is common for most self-supervised methods—since there is a reduced amount of available information in the dataset. Table 11 show these results in the revised manuscript (a copy of which is included below).
>
> | Modality                   | Methods    | Epoch duration (s) | Epochs |
> |:-------------------------- |:---------- |:------------------:|:------:|
> | Image inpainting           | Supervised |         12         |  200   |
> |                            | ES (Ours)  |         12         |  1000  |
> |                            | EI         |         14         |  1000  |
> | MRI (x8 Accel., 40 dB SNR) | Supervised |         29         |  200   |
> |                            | ES (Ours)  |         24         | 13800  |
> |                            | EI         |         53         |  7800  |
> |                            | SURE       |         36         |  5700  |
> | MRI (x6 Accel., 10 dB SNR) | Supervised |         19         |   70   |
> |                            | ES (Ours)  |         19         |  3100  |
> |                            | EI         |         75         |  2400  |
> |                            | SURE       |         35         |  1200  |
>
> **Table 11. Training durations.** For each training, we report the average epoch duration and the number of epochs until the model is trained. Inpainting trainings are conducted on a single NVIDIA RTX 3090 Ti GPU and MRI trainings on a NVIDIA H100 GPU.

---

> ### Author Response · Authors · 2025-11-21
> **Response part 4**
>
> > Computational Overhead and Scalability Concerns
>     - The use of Reynolds averaging in Eq. 18 for enforcing equivariance becomes prohibitively expensive for large groups (e.g., continuous shifts or rotations), as noted in Appendix A. The paper only evaluates small discrete groups (e.g., 90° rotations + flips), limiting generalizability.
>
> We would like to point out that one of *the main contribution of the paper is to show that Reynolds averaging is not necessary if an equivariant reconstruction network is used*. This is the case for the shift-equivariant networks used in the compressed sensing and inpainting experiments. The result also holds for continuous translations (with infinitely many group elements) as layer-wise equivariant network architectures can be used instead (Karras et al., 2021; Michaeli et al., 2023).
>
> For rotations, we observed that the existing layer-wise architectures (Cohen and Welling, 2017; Weiler and Cesa, 2019) perform worse than Reynolds' averaging in our setting, and we thus included results with averaging. Nonetheless, we expect that our method will benefit from the next advances in the field of equivariant architectures.
>
> - Steerable CNNs, Cohen and Welling, ICLR 2017
> - General E(2) - Equivariant Steerable CNNs, Cohen and Cesa, NeurIPS 2019
> - Alias-Free Generative Adversarial Networks, Karras et al., NeurIPS 2021
> - Alias-Free Convnets: Fractional Shift Invariance via Polynomial Activations, Michaeli et al., CVPR 2023
>
> > Corollary 1 states that the forward operator A must not be equivariant w.r.t. $G$ for $Q_{A_1}$ to be full rank. In practice, how should one select $G$ given a fixed $A$? Are there automated or data-driven strategies to choose transformations that maximize the rank of $Q_{A_1}$?
>
> We believe that developing automated or data-driven strategies for the selection of transformations for equivariance-based self-supervised imaging algorithms is an interesting research direction, but it is outside the scope of this work. Instead, we have added a section in Appendix A.2 discussing how the transforms should be chosen in practice, together with a table listing correct choices of transformations for different measurement operators (Table 5). See below for a copy of Appendix A.2 and Table 5.
>
> _There are two major criteria for choosing the transformations for a given application. First, the image distribution of interest should be invariant to the chosen transformations. Aerial, remote sensing and microscopic images are invariant to translations and rotations as the scenes and subjects they measure exhibit no privileged position and orientation with respect to the image plane. Natural image distributions and texture distributions (Portilla & Simoncelli, 2000) are also generally invariant to translations, but they are less invariant to rotations, as natural images are typically oriented upward and texture distributions might be anisotropic._
>
> _Second, the transformations should also be chosen in accordance with the measurement operator. Corollary 1 shows that transformations for which the operator is equivariant do not improve the reconstruction process. It is a well-known criterion introduced in the original work on equivariant imaging (Chen et al., 2021), which remains correct in our setting where measurement splitting is added to the theoretical analysis. Table 5 lists correct choices of transformations for common measurement operators._
>
>
> | Operator            | Translation | Rotation | Permutation | Amplitude |
> | ------------------- | ----------- | -------- | ----------- | --------- |
> | Isotropic blur      | ✗           | ✗        | ✓           | ✗         |
> | Image inpainting    | ✓           | ✓        | ✓           | ✗         |
> | Sparse-view CT      | ✗           | ✓        | ✓           | ✗         |
> | Accelerated MRI     | ✗           | ✓        | ✓           | ✗         |
> | Compressive sensing | ✓           | ✓        | ✓           | ✗         |
>
> **Table 5: Decision table for the transformations.** Corollary 1 shows that not all transformations are well-suited for all problems. Namely, transformations for which the operator is equivariant introduce no additional information for learning in the nullspace of the operator. This table specifies which transformations are well-suited for which operator.

---

> > ### Comment · Reviewer_ZYHS · 2025-11-27
> >
> > I thank the reviewers for their effort to answer my questions and try to address some of my main concerns. Still, I believe that while there is definitely merit in this work, the experimental evaluation of the proposed strategy is rather weak and relies on simplified problems that are very far from those related to real-world applications. The newly added experiment on evenly-spaced masking instead of random masking, clearly indicates that the drop in performance compared to supervised methods is rather large. Also it is my opinion that using the term image inpainting for these type of tasks is somehow misleading, since typically this term refers to the infilling of relatively large unified (connected) areas.
> >
> > For the above reasons I am retaining my original score.

---

> > > ### Author Response · Authors · 2025-12-03
> > >
> > > Thank you for your comments.
> > >
> > > We consider it unfair to penalize our work because a choice of transformations (which can be seen as hyperparameters of our algorithm) that we advocate against leads to subpar performance. It is at your request that we evaluate the use of shifts for inpainting with evenly-spaced push-broom masks. Indeed, it is a successful verification of Corollary 1 and **not** a failure case of our method. The choice of transformations is one of the most important hyperparameters of our method and we make an extensive case for it in the manuscript. We demonstrate in this specific experiment that a poor choice of transformations leads to low performance whereas in all other experiments, the choice of transformations is made in accordance with the criteria listed in the manuscript, which leads to high performance.
> > >
> > > We agree that our method could be evaluated in more realistic settings. We believe however that the level of realism of our experiments is largely sufficient to meet the highest standards in the challenging field of self-supervised imaging and to sustain the claims we make throughout the work. In particular, the use of synthetic measurements can be replaced with a pre-processing pipeline carefully designed to account for the specific processing of the raw measurement data made available to the practitioner. While important, it is largely orthogonal to the challenges we set to overcome in this work. We nonetheless update the manuscript with an additional experiment where our method is evaluated on raw MRI data instead of synthetic measurements. The updated Table 1 (copied below) shows that the performance of our method is expectedly not hindered by this change.
> > >
> > > We also emphasize that the randomly-spaced push-broom masks in our experiments are, in particular, a realistic model of LinoSPAD data acquisition (Lindell et al., 2018).
> > >
> > > We also include an experiment on an additional modality to further demonstrate the broad applicability of our method. Namely, we evaluate our method on sparse-view computed tomography (CT) using thoracic scans from the LIDC-IDRI dataset (Armato III et al., 2011). The updated Table 1 (copied below) shows that our method performs competitively on this modality as well.
> > >
> > > | Modality                          | Method     |    PSNR $\uparrow$   |    SSIM $\uparrow$      |    EQUIV $\uparrow$  |
> > > |:--------------------------------- |:---------- |:--------------------:|:-----------------------:|:--------------------:|
> > > | MRI (x8 Accel., 40 dB SNR)        | Supervised | 28.74          | 0.6445          | 31.71          |
> > > |                                   | ES (Ours)  | **28.54** | **0.6195** | **31.53** |
> > > |                                   | EI         | 27.88          | 0.5731          | 30.79          |
> > > |                                   | SURE       | 24.45          | 0.5479          | 27.35          |
> > > |                                   | IDFT       | 23.62          | 0.5052          | 25.99          |
> > > | Real MRI measurements (x8 Accel.) | Supervised | 28.81          | 0.6480          | 31.81          |
> > > |                                   | ES (Ours)  | **28.30** | **0.6151** | **31.29** |
> > > |                                   | EI         | 27.88          | 0.5740          | 30.80          |
> > > |                                   | MC         | 23.63          | 0.5061          | 26.00          |
> > > |                                   | IDFT       | 23.63          | 0.5060          | 26.00          |
> > > | CT (50 views, 50 dB SNR)          | Supervised | 33.99          | 0.8819          | 34.00          |
> > > |                                   | ES (Ours)  | **32.62** | **0.8570** | **32.60** |
> > > |                                   | EI         | 28.61          | 0.7400          | 28.61          |
> > > |                                   | FBP        | 25.59          | 0.4805          | 25.59          |
> > >
> > > **Table 1. Medical imaging results.** ES (ours) performs better than EI, SURE and MC (baselines), while performing almost as well as the supervised baseline in reconstruction quality (PSNR, SSIM) and measured equivariance (EQUIV). In **bold**, the best self-supervised metrics (avg $\pm$ st.d.).
> > >
> > > - Armato III, S. G., McLennan, G., Bidaut, L., McNitt‐Gray, M. F., Meyer, C. R., Reeves, A. P., ... & Clarke, L. P. (2011). The lung image database consortium (LIDC) and image database resource initiative (IDRI): a completed reference database of lung nodules on CT scans. Medical physics, 38(2), 915-931.
> > > - Lindell, D. B., O'Toole, M., & Wetzstein, G. (2018). Single-photon 3d imaging with deep sensor fusion. ACM Trans. Graph., 37(4), 113.

---

### Official Review · Reviewer_zSdB · 2025-11-02

**Soundness:** 1
**Presentation:** 2
**Contribution:** 1
**Rating:** 2
**Confidence:** 4

**Summary:**

This paper discusses a study of self-supervised method to solve inverse problems, with the approach as designing loss functions with the "SPLIT" technique.

**Strengths:**

Structure of the paper is clear.

**Weaknesses:**

- Theorem 1 is well expected. If $(AT_g)^\top AT_g$ has full rank, then $AT_g$ itself does not dimensionally compress information, hence the ill-conditioning of the inverse problem is no-longer a concern.

- Throughout this paper, the notation is largely unclear.

    - It is hence by requested that authors specify dimension for all matrix quantities for clarity.
    - With expect to all expectations, authors explicitly explain which quantities are being integrated with respect to its distribution and with respect to which quantities is the variable still stochastic.

- In the experiment section, authors should very clearly define what the measurement matrix $A$ is for all scenarios. Explicitly explain its shape and why the measurement is a compressive measurement.
    - Because of the proposed "SPLIT" technique, authors should also explain clearly what $A_2$ is in every testing scenario.

- In the experiment section, authors should clearly list the citation of used baselines, and list the definition of all metrics, in the main text or in the appendix. Say, what is "EI"?

-  In the methodology section, authors should state what the final proposed algorithm is, in a crystal clear way.

**Questions:**

- What does $\mathbb{E}_{x | y, A} \{ x \}$ mean, exactly? Do you mean the reconstructed $\widehat{x}$ given measurement matrix and measurement observation $y$?
- What does $A_1 | y, AT_g$ mean, exactly?
- In section 5.1, multiplying by "measurement matrix with fewer columns than rows" is not compressive, is it?

---

> ### Author Response · Authors · 2025-11-21
> **Response part 1**
>
> Thank you for your comments and suggestions. We address them below and in the revised manuscript with the changes in red.
>
> > Theorem 1 is well expected. If $(AT_g)^\top AT_g$ has full rank, then $AT_g$ itself does not dimensionally compress information, hence the ill-conditioning of the inverse problem is no longer a concern.
>
> There appears to be a confusion about the statement of Theorem 1. The matrix that should have full rank for the condition of the theorem to be verified is not $(AT_g)^\top AT_g$, but rather
> $$
> \mathbb E_{g \ | \ A_1}\Big[(A T_g)^\top A T_g\Big] := \frac{1}{|G|}\ \sum_{g=1}^{|G|} (A T_g)^\top \ A T_g\ p(g \ | \ A_1).
> $$
> This matrix can have full rank even when $(A T_g)^\top A T_g$ is rank deficient for all $g\in \{1,\dots,|G|\}$. In particular, **in all our experiments (MRI, compressed sensing and inpainting), the matrix $(AT_g)^\top AT_g$ does not have full rank**. Our theorem constitutes a new contribution since it can handle the rank-deficient case.
>
> For example, consider an inpainting problem in dimension $n=3$ with $G$ the group of (circular) shifts in $\mathbb{R}^3$ ($|G|=3$), where we observe $m=2$ pixels (out of the 3 pixels). In this case, the matrices are:
> $$A = AT_0 = \begin{bmatrix} 1 & 0 & 0 \\\ 0 & 1 & 0 \end{bmatrix}, \
> AT_1 = \begin{bmatrix} 1 & 0 & 0  \\\ 0 & 0& 1 \end{bmatrix}, \
> AT_2 = \begin{bmatrix} 0 & 1 & 0 \\\ 0 & 0& 1 \end{bmatrix}. $$
> Conditioning the expectation on the splitting $A_1= \begin{bmatrix} 1 & 0 & 0 \end{bmatrix} \in \mathbb{R}^{1\times 3}$ of $A$, we have
> $$\mathbb{E}_{g | A_1}[(AT_g)^\top AT_g] = \frac{1}{2}(AT_0)^\top AT_0 + \frac{1}{2}(AT_1)^\top AT_1 = \begin{bmatrix} 1 & 0 & 0 \\\ 0 & 1/2 & 0 \\\ 0 & 0& 1/2 \end{bmatrix}.$$
>
> > Throughout this paper, the notation is largely unclear.
>     - It is hence by requested that authors specify dimension for all matrix quantities for clarity.
>
> Thank you for suggesting this. The dimension of $A \in \mathbb R^{m \times n}$ is given in the introduction. We have added the dimension of $T_g \in \mathbb R^{n \times n}$ in the updated manuscript (see Section 3.2). We have also added the dimensions of $\mathbf A_1 \in \mathbb R^{m_1 \times n}$, $ A_2 \in \mathbb R^{m_2 \times n}$, $Q_{A_1} \in \mathbb R^{n \times n}$ and $\bar Q_{A_1} \in \mathbb R^{n \times n}$ where they are defined in the revised manuscript.
>
> > Throughout this paper, the notation is largely unclear.
>     - With expect to all expectations, authors explicitly explain which quantities are being integrated with respect to its distribution and with respect to which quantities is the variable still stochastic.
> > What does $E_{x|y,A} x$ mean, exactly? Do you mean the reconstructed $\hat{x}$ given measurement matrix and measurement observation $y$?
>
> We followed the notation typically used in self-supervised literature (Pang et al., 2021; Monroy et al., 2025), where $\mathbb E_{\mathbf{x} \mid \mathbf{y}, \mathbf{A}}\{\mathbf{x}\}=\int_\mathbf{x} \mathbf{x}p(\mathbf{x} \mid \mathbf{y}, \mathbf{A})\mathrm{d}\mathbf{x}$ denotes the expectation of $x$ with respect to $p(\mathbf{x} \mid \mathbf{y}, \mathbf{A})$, the distribution of $x$ conditioned on knowing $\mathbf{y}$ and $\mathbf{A}$. In other words, $\mathbb{E}_{\mathbf{x} \mid \mathbf{y}, \mathbf{A}}\{\mathbf{x}\}$ is the expectation of all $\mathbf{x}$ such that $\mathbf{y} =\mathbf{A} \mathbf{x}$ (or in noisy measurement case: $\mathbf{y} =\mathbf{A} \mathbf{x} + \mathbf{\varepsilon}$) for a given $\mathbf{y}$ and $\mathbf{A}$.
>
> - Pang, T., Zheng, H., Quan, Y. and Ji, H., 2021. Recorrupted-to-recorrupted: Unsupervised deep learning for image denoising. In Proceedings of the IEEE/CVF conference on computer vision and pattern recognition (pp. 2043-2052).
> - Monroy, B., Bacca, J. and Tachella, J., 2025. Generalized recorrupted-to-recorrupted: Self-supervised learning beyond gaussian noise. In Proceedings of the Computer Vision and Pattern Recognition Conference (pp. 28155-28164).
>
> > What does $A_1 | y, AT_g$ mean, exactly?
>
> The notation $\mathbf A_1 \mid \mathbf{y}, \mathbf{A}\mathbf T_g$ is not used in the paper. However, we do have similar notation, such as $\mathbb E_{\mathbf y_1, \mathbf{A}_1 \mid \mathbf{y}, \mathbf{A}\mathbf{T}_g}\{ \cdots \}$ in equation (8) of the paper. This expectation means that we integrate with respect to the joint distribution of $\mathbf{y}_1$ and $\mathbf{A}_1$ conditioned on knowing $\mathbf{y}$ and $\mathbf{A}\mathbf{T}_g$. We are thus considering all the couples $(\mathbf{y}_1, \mathbf{A}_1)$ which might have originated by splitting the known vector $\mathbf{y}$ and matrix $\mathbf{A}\mathbf{T}_g$.
>
> We have added in the manuscript (page 3) a footnote stating: We use the notation $\mathbb E_{a\mid b} \left\\{g(a)\right\\}$ for $\int_{a} g(a)p(a|b)da$

---

> ### Author Response · Authors · 2025-11-21
> **Response 2**
>
> > In the experiment section, authors should very clearly define what the measurement matrix is for all scenarios. Explicitly explain its shape and why the measurement is a compressive measurement.
> >  - Because of the proposed "SPLIT" technique, authors should also explain clearly what $A_2$ is in every testing scenario.
>
>
> We have updated the experiment section to clearly define the measurement matrix $A$ for all scenarios, including its dimensions.  We explicitly specify $A_1$ and $A_2$, and state that they are two complementary sub-matrices of $A$ of shape $m_1$ and $m_2$ with $m_1 + m_2 = m$.
> For all experiments and scenarios, we used $m_1 = 0.8 m$ (and so $m_2 = 0.2 m$). This information is included in Appendix B.3.
>
> We detail the matrix $A$ considered for each scenario:
> - **Compressive sensing:** The forward operator $A \in \mathbb{R}^{m\times n}$ is a random Gaussian matrix with $m<n$, i.e. all entries are i.i.d Gaussian variables: $\forall i,j, \ a_{i,j} \sim \mathcal{N}(0, \frac{1}{m})$.
> For this experiment, we have $n=784$ and $m$ varies in $\{0.5n, 0.4n, 0.3n, 0.2n, 0.1n\}$.
>
> - **Image inpainting:** The forward operator  $A \in \mathbb{R}^{m\times n}$ is a random binary mask. This matrix is obtained by sampling $a_1, \ldots, a_n \sim  \text{Ber}(0.3)$ where $n = 128 \times 128$ and letting $A_{i,j} = \delta_{j_i,j}$ for $i = 1, \ldots, m$ and $j = 1, \ldots, n$, where $m \approx 0.3n$ is the number of nonzero values in $a$, and $j_i$ is the $i$-th index in $a$ corresponding to a nonzero value. The measurements thus obtained are compressed measurements with probability $1 - (0.3)^n$. The images have three channels (RGB) resulting in $A$ being the the same inpainting mask concatenated three times $A \leftarrow \text{diag}(A, A, A) \in \mathbb{R}^{3m \times 3n}$. In this scenario $n=3\times 128 \times 128 =49152$ and $m\approx 0.3 \times 49152 =14745$.
>
> - **MRI:** The forward operator $A \in \mathbb R^{m \times n}$ is expressed as $A =MF$ where $F \in \mathbb R^{n \times n}$ denotes the $n \times n$ discrete Fourier transform matrix and where $M \in \mathbb R^{m \times n}$ is the subsampling mask defined as $M_{i,j} = \delta_{j_i,j}$ for $i = 1, \ldots, m$ and $j = 1, \ldots, n$, where $j_i$ denotes the $i$-th component in $\mathbb R^n$ corresponding to a pixel in one of the subsampled vertical lines in a random Gaussian mask. For FastMRI we have $n=320\times 320=102400$ and $m = 0.5\times 102400 = 51200$.
>
> > In the experiment section, authors should clearly list the citation of used baselines, and list the definition of all metrics, in the main text or in the appendix. Say, what is "EI"?
>
> We have moved the definition of the acronym EI from the introduction (page 2, line 58) to the background section (page 4, line 169). We have also added the reference a second time, in the experimental section, where all baselines are listed. We hope this simplifies searching for the papers associated with each baseline.
>
> We have also included definitions of the 3 metrics used in the paper in Appendix B.4 of the updated manuscript.
>
> > In the methodology section, authors should state what the final proposed algorithm is, in a crystal clear way.
>
> The method proposed in this paper consists of minimizing the loss presented in Eq. 7, where the expectation on $A_1$ and $y_1$ is approximated in practice by sampling a random split $(y_1,A_1)$ per minibatch. This explanation was already present in lines 216-244 of the paper.
>
> Following your suggestion, we have also included pseudo-code for the training and testing phases, in the hope that this will improve the understanding of the paper. Please see Algorithm 1 and Algorithm 2 in Appendix A.3.
>
> > In section 5.1, multiplying by "measurement matrix with fewer columns than rows" is not compressive, is it?
>
> Thank you for pointing out this typo. We have corrected it to ‘fewer rows than columns’.

---

### Official Review · Reviewer_46Uu · 2025-11-02

**Soundness:** 4
**Presentation:** 3
**Contribution:** 3
**Rating:** 6
**Confidence:** 4

**Summary:**

This paper addresses self-supervised learning for inverse problems in settings where practitioners lack access to ground truth data and must train from incomplete measurements. The proposed  approach relies on the assumption that the ground truth distribution is invariant to a group of transformations, as formalized in Assumption 1. Building on this assumption, the authors propose a novel self-supervised loss function, Equivariant Splitting (ES), which trains the reconstruction network by optimizing consistency between full and partial measurements, averaged over symmetry transformations of the forward operator. The authors provide theoretical justification by showing that their expected loss converges to the MMSE estimator under certain conditions. Furthermore, they demonstrate empirically that their proposed method achieves strong performance on natural images and MRI reconstruction across a diverse range of inverse problems.

**Strengths:**

**Strengths:**

- The paper presents a novel method that addresses a more realistic setting than traditional generative prior approaches, which typically assume access to ground truth images or paired ground truth-measurement data. The combination of equivariant priors with measurement splitting is original and well-motivated.

- The authors provide theoretical analysis demonstrating that the proposed loss converges in expectation to the MMSE estimator under certain conditions, offering solid justification for the approach.

- The empirical results are strong, achieving performance competitive with supervised baselines that have access to ground truth data, which validates the practical effectiveness of the self-supervised approach.

**Weaknesses:**

**Weaknesses:**

- Assumption 1 would benefit from additional discussion regarding the practical scope of applicable transformations for different problem domains. While the general framework is elegant, natural images often exhibit partial rather than full symmetries (e.g., small rotations of 15° are reasonable, while 180° rotations may be implausible). Providing examples for specific domains would strengthen the paper and provide valuable guidance for practitioners applying this method to new problems.


- The theoretical analysis provides useful motivation for the proposed method, though the mathematical results could be presented differently to better reflect their contribution. The proofs largely build on established techniques. This is a significant portion of the paper.

- Mirror Weakness Theorem 2 makes interesting claims about Reynolds averaging and MAP estimates that would benefit from empirical validation. To my knowledge I did not see anything.

**Questions:**

**Questions:**

1. The equivariance framework is quite general and could potentially apply to many problem domains beyond those explored in the paper. Could the authors provide additional guidance on selecting appropriate transformation groups for different applications? For instance, a discussion of which symmetries are valid for medical imaging, computational photography, or scientific imaging would be valuable for practitioners. Understanding the broader applicability and potential impact of this framework across different domains would strengthen the paper's contribution.

2. The theoretical analysis provides helpful motivation for the proposed method. Could the authors clarify which aspects of the proof techniques represent novel contributions versus adaptations of existing results?

---

> ### Author Response · Authors · 2025-11-21
> **Response part 1**
>
> Thank you for your comments and suggestions. We address them below and in the revised manuscript with the changes in red.
>
> > Assumption 1 would benefit from additional discussion regarding the practical scope of applicable transformations for different problem domains. While the general framework is elegant, natural images often exhibit partial rather than full symmetries (e.g., small rotations of 15° are reasonable, while 180° rotations may be implausible). Providing examples for specific domains would strengthen the paper and provide valuable guidance for practitioners applying this method to new problems.
>
> Thanks for this suggestion. We have added the following paragraph to clarify the broad applicability of Assumption 1:
>
> _This assumption applies in many different settings, as natural image distributions are generally invariant to rigid transformations (translations, rotations, flips), especially aerial and remote sensing images which have no privileged orientation. Moreover, our experiments show that our experiment performs well even in the presence of approximate invariance, e.g., for medical images._
>
> Notably, our experiments use **real-world image datasets that are not perfectly invariant** (nor have they been made invariant in a synthetic way, as often done in equivariance-related papers). Most digits in MNIST and objects in DIV2K scenes are located at the center of the image, and most knee scans in FastMRI share a similar vertical orientation. Despite the datasets not being exactly invariant, our experiments show that the proposed loss remains effective at learning from data associated with a single rank-deficient operator.
>
> > The theoretical analysis provides useful motivation for the proposed method, though the mathematical results could be presented differently to better reflect their contribution. The proofs largely build on established techniques. This is a significant portion of the paper.
>
> We prefer to leave formal statements in the main text and detailed proofs in the appendix for the sake of clarity. Could you please clarify what do you mean by "the mathematical results could be presented differently"?
>
> > Mirror Weakness Theorem 2 makes interesting claims about Reynolds averaging and MAP estimates that would benefit from empirical validation. To my knowledge I did not see anything.
>
> We have added an empirical verification of the equivariance of MAP estimators in the appendix:
>
> _We verify empirically that MAP reconstructors are equivariant as long as the prior is itself equivariant, i.e., the claim made in Theorem 2. Since they cannot be computed exactly in general, we consider a specific scenario where the computation is available in closed form. We assume that 1) $p(x)$ is given by $\mathcal{N}(0, \tau^2 I_n)$, 2) $A \in \mathbb R^{m \times n}$ is the two-dimensional decimation operator with decimation rate $2$, 3) $y \mid Ax \sim \mathcal N(0, \sigma^2 I_m)$, and 4) that $T_g$ denotes the rotation by angle $g \in \{ 0^{\circ}, 90^{\circ}, 180^{\circ}, 270^{\circ} \}$. Under these assumptions, the prior distribution is equivariant and the MAP estimator in eq. (23) can be expressed in closed-form as
> $$
>         f(\mathbf{y}, \mathbf{A} \mathbf T_g) = \frac{\tau^2}{\tau^2 + \sigma^2} \mathbf T_g^{-1} \mathbf{A}^\top \mathbf{y}
> $$
> with $f(\mathbf{y}, \mathbf{A})$ being the special case where $T_g = I_n$. In the experiment, we set $n = 128 \times 128$ for a grayscale image with $128$ rows and $128$ columns and we compute the equivariance metric in eq. (19) (EQUIV) for the MAP reconstructor using 256 i.i.d. samples from the joint distribution. Table 9 shows the results for every angle and for the average over all angles. As predicted theoretically, perfect equivariance is achieved._
>
> |           | 0°       | 90°      | 180°     | 270° | Average  |
> | --------- | -------- | -------- | -------- | ---- | -------- |
> | **EQUIV** | $\infty$ | $\infty$ | $\infty$ | $\infty$    | $\infty$ |
>
> **Table 9. Empirical validation of the equivariance of MAP estimators.** Infinity values indicate a perfect match up to machine precision.

---

> ### Author Response · Authors · 2025-11-21
> **Response part 2**
>
> > The equivariance framework is quite general and could potentially apply to many problem domains beyond those explored in the paper. Could the authors provide additional guidance on selecting appropriate transformation groups for different applications? For instance, a discussion of which symmetries are valid for medical imaging, computational photography, or scientific imaging would be valuable for practitioners. Understanding the broader applicability and potential impact of this framework across different domains would strengthen the paper's contribution.
>
> The choice of transformations is conditioned by two criteria. The first criterion is that the transformations should be chosen so that the image distribution of interest is invariant to them. Aerial, remote sensing and microscopic images are invariant to translations and rotations as the scenes and subjects they measure exhibit no privileged position and orientation with respect to the image plane. Natural image distributions and texture distributions (Portilla et al. 2000) are also generally invariant to translations, but they are less invariant to rotations as natural images are typically oriented upward and texture distributions might be anisotropic.
>
> It is also important that the transformation is chosen in accordance with the measurement operator. Corollary 1 shows that transformations for which the operator is equivariant introduce no additional information that could help in the reconstruction process. It is a well-known criterion introduced in the original work on equivariant imaging (Chen et al., 2021), which remains correct in our setting where measurement splitting is added to the theoretical analysis. We have added this explanation as a decision aid in Appendix A.2 and a table listing correct choices of transformations for different operators (Table 5). See below for a copy of Table 5.
>
>
> | Operator             | Translation | Rotation | Permutation | Amplitude |
> |----------------------|-------------|----------|-------------|-----------|
> | Isotropic blur        | ✗           | ✗        | ✓           | ✗         |
> | Image inpainting     | ✓           | ✓        | ✓           | ✗         |
> | Sparse-view CT       | ✗           | ✓        | ✓           | ✗         |
> | Accelerated MRI      | ✗           | ✓        | ✓           | ✗         |
> | Compressive sensing  | ✓           | ✓        | ✓           | ✗         |
>
> **Table 5: Decision table for the transformations.** Corollary 1 shows that not all transformations are well-suited for all problems. Namely, transformations for which the operator is equivariant introduce no additional information for learning in the nullspace of the operator. This table specifies which transformations are well-suited for which operator.
>
> - Portilla, J., & Simoncelli, E. P. (2000). A parametric texture model based on joint statistics of complex wavelet coefficients. International journal of computer vision, 40(1), 49-70.
> - Chen, D., Tachella, J. and Davies, M.E., 2021. Equivariant imaging: Learning beyond the range space. In Proceedings of the IEEE/CVF International Conference on Computer Vision (pp. 4379-4388).
>
> > The theoretical analysis provides helpful motivation for the proposed method. Could the authors clarify which aspects of the proof techniques represent novel contributions versus adaptations of existing results?
>
> We believe the conditioning steps in the proof of Theorem 1 constitute an important contribution in terms of proof techniques, since they require a good understanding of the interplay between the group action and the random splitting. Moreover, we believe our definition of equivariant reconstruction functions (Definition 1), despite being relatively simple and easy to understand, did not exist in previous literature, and will serve as a stepping stone for future theoretical works involving equivariance in inverse problems.

---

### Author Response · Authors · 2025-12-03
**Message to the area chair (1/2)**

**Message to the area chair**

Thank you for reviewing our submission and for the extra work due to this unexpected situation.

In an attempt to accommodate with the disrupted reviewing process, we made the decision to include below a short summary of the reviewers' comments and concerns and how we addressed them. The changes are displayed in red and purple in the revised manuscript. Changes in red have been introduced after the first round of review and changes in purple after the second.

Experiments added in the rebuttal
---------------------------------

We have conducted additional experiments to address the reviewers' comments:

* **Push-broom inpainting.** We conducted two experiments simulating "push-broom" corruptions by removing entire image columns. The first scenario involves randomly selected columns, while the second removes every other column.
* **Real measurements MRI.** We evaluate our method on real FastMRI measurements in addition to our experiments with synthetic measurements.
* **Computed Tomography (CT).** We evaluate our method on sparse-view computed tomography in addition to compressive sensing, inpainting and accelerated MRI.
* **Sensitivity to noise distribution shifts.** We evaluate the sensitivity of our method to mis-estimated noise levels across various imaging modalities.

Summary of Reviewer concerns and how we addressed them
------------------------------------------------------

**Reviewer 46Uu** (score: 6/10) points out that the work would benefit from a statement clarifying the applicability of Assumption 1 to practical image distributions with partial and/or inexact invariances. They suggest that the mathematical results, while relevant, could be presented differently to highlight their novelty better. They also request an empirical validation of Theorem 2 and the addition of further guidance on the choice of transformations for the different domains of application (medical imaging, photography, etc.).

We have clarified that our method is applicable to real-world settings by noting that *our experiments on DIV2K and FastMRI succeed despite these datasets not being perfectly invariant*. To address the theoretical concerns, we have added an empirical verification (Table 9) confirming the equivariance of MAP estimators in a controlled setting. We have also highlighted the novelty of our proof techniques, specifically the conditioning steps in Theorem 1. Finally, we provided a new decision table (Table 5) and added in Appendix A.2 a guide to practitioners in choosing transformations.

Unfortunately, the reviewer did not have enough time to re-evaluate the paper with these clarifications, which could have raised their original score, since the reviewer i) had given "excellent" soundness to the method, ii) considered our approach as being **"a novel method that addresses a more realistic setting than traditional generative prior approaches"** with a **"theoretical analysis (...) offering solid justification"**, and iii) stating that **"empirical results are strong"**.

---------------
**Reviewer zSdB** (score: 2/10) questions the non-triviality of Theorem 1, arguing that if the matrix product $(AT_g)^\top AT_g$ has full rank, there is no compression, making the theorem expected. They also criticize the notation, finding dimensions and expectation definitions unclear. Furthermore, they requested precise definitions for the measurement matrices $A$ and metrics used in the experiments, and asked for a "crystal-clear" statement or pseudocode of the proposed algorithm.

We address the reviewer's misunderstanding of Theorem 1 with a precise explanation that it applies if the averaged matrix has full rank, which is distinct from every matrix having full rank itself. This is crucially what enables the method to be applied successfully to problems with a rank-deficient forward matrix. To improve the clarity of the work, we have completed the manuscript with condensed algorithms summarizing our method, the dimensions of the matrices we consider, a definition of our notation for conditional expectations, mathematical descriptions of the forward matrices in the experiments and additional details about the performance metrics.

We respectfully believe that the reviewer is unfamiliar with some of the mathematical notation in the paper (e.g. the conditional expectation $\mathbb{E}_{x|y,A}\{x\}$), despite being the standard notation in this field - see references in the discussion. **We believe that the low score (2/10) stems primarily from this misunderstanding of mathematical notation**. Unfortunately, the reviewer did not have enough time to consider the clarifications we provided in the rebuttal and revised manuscript, preventing a proper re-evaluation of our work.

(...)

---

> ### Author Response · Authors · 2025-12-03
> **Message to the area chair (2/2)**
>
> (...)
>
> **Reviewer ZYHS** (score: 4/10) raises concerns about the applicability of Assumption 1 when it is only approximately met, and our use of the same implementation of the forward models to constitute the datasets. They question whether our use of group-averaging to obtain equivariance to rotations and flips prevents our method from scaling well. They also find the experiments to be insufficiently realistic and they request additional guidance on the choice of transformations for the application of our method.
>
> We studied the method's behavior when invariance assumptions are violated by adding "Push-broom" experiments (Table 7), which show that the method holds up under approximate equivariance but fails if the operator is fully equivariant (as predicted by our theoretical framework). To address realism, we added an ablation study (Table 8) demonstrating robustness to unknown noise and introduced a more realistic satellite imaging scenario. We clarified that Monte Carlo sampling keeps per-epoch costs low (Table 11) and noted that Reynolds averaging is avoidable with equivariant architectures (which constitutes a main contribution of our paper, see Theorem 3). Finally, we added Appendix A.2 and Table 5 to guide the selection of transformations based on operator and dataset properties.
>
> We believe that our rebuttal would have prompted the reviewer to re-evaluate their score as **it addresses their 3 main concerns**. Namely, i) we have performed experiments on real datasets that are not perfectly invariant, ii) we have added results on real fastMRI measurements showing good performance even though the noise distribution is unknown, and iii) we have clarified that the computational cost of our method is comparable to or better than other self-supervised baselines.
>
> ---------------
> **Reviewer vQFi** (score: 4/10) requests a quantitative analysis of performance when the assumption of invariance holds only approximately. They also request additional comparisons with diffusion models (DDRM), and the release of the code and random seeds used in the experiments for reproducibility. They question the level of realism of the experiments and they find the practical recommendations for choosing the transformations to be insufficiently complete. Finally, they ask how to verify the rank condition of Theorem 1 in practice.
>
> We have directed the reviewer to Table 5 for practical guidelines on selecting the transformations based on operator properties. Regarding the rank condition, we explained that while explicit computation is difficult, the spectrum can be approximated using randomized linear algebra techniques. We clarified that diffusion models like DDRM fall outside the scope of this work as they are not self-supervised baselines given they require a supervised pre-training. To address the realism concerns, we added a more realistic Push-broom experiment. Finally, we ensured reproducibility by providing a link to an anonymous repository containing the entire code, seeds, and reproduction scripts, and we updated the text to specify training details such as the number of splits used.
>
> We believe that our responses to the 5 questions of the reviewer, our clarification stating that good experimental results were obtained without exact equivariance (using unaltered popular datasets which are not perfectly invariant), as well as making the code available for reproducibility, might have convinced the reviewer to reconsider their rating. Indeed, they found the paper to provide a **"unifying view that connects splitting losses and equivariant imaging and states the assumptions under which ES is theoretically unbiased for the supervised objective"**, introduce the "definition of equivariant reconstructors" and enumerate "common architectures that satisfy it", that **"the method applicable to realistic measurement noise models"** and "Experiments span multiple inverse problems" and "demonstrate the claimed synergy between splitting losses and equivariant architectures".

---

### Meta-Review · Area_Chair_bWUX · 2026-01-02

**Summary:**

The reviewers generally agree that the paper addresses an important and challenging setting and that the proposed Equivariant Splitting (ES) method is technically sound and empirically competitive. However, the reviews appear somewhat misaligned in terms of what is being evaluated as the main contribution.

Several weaknesses raised by the reviewers (e.g., strong invariance assumptions, dependence on equivariance, idealized operators, and limitations of Reynolds averaging) are largely inherited from prior Equivariant Imaging (EI) work, rather than being specific shortcomings of ES itself. As a result, much of the criticism focuses on the general limitations of equivariance-based self-supervision, without carefully assessing whether ES provides a meaningful conceptual or technical advance relative to EI.

At the same time, the novelty question, namely, whether combining measurement splitting with EI-style equivariance constitutes a sufficiently new contribution, and whether the resulting theoretical analysis (e.g., unbiasedness of the loss under equivariance) represents a genuine improvement over EI, has not been consistently or deeply examined across reviews. Some reviewers view ES as a unifying or extending framework, while others implicitly treat it as incremental, but this discrepancy is not clearly resolved.

Given the current set of reviews and scores (ranging from reject to weak accept, 6/4/4/2), it is difficult to make a definitive accept/reject decision without further discussion. In particular, additional clarification is needed on (i) how ES substantively improves upon EI beyond reusing its assumptions, and (ii) whether this improvement rises to the level of novelty expected for acceptance. A focused discussion on the relationship between ES and EI would be valuable for a fair final decision.

**Reviewer Concerns:**

Concerns addressed:
The rebuttal addressed several clarification-level concerns raised by the reviewers, including notation issues, clearer specification of the measurement operators and splits used in experiments, and additional explanations regarding Assumption 1 and the practical choice of transformation groups. These points largely resolve presentation-related criticisms and improve the readability of the paper. The authors also strengthened the empirical validation in response to the reviewers’ concerns about realism and the scope of experiments.

Concerns still outstanding:
Some concerns regarding the strength of the theoretical assumptions (e.g., invariance) remain largely conceptual and were not fundamentally changed by the rebuttal. These assumptions are intrinsic to the proposed framework and similar equivariance-based approaches.

**Reviewer Scores:**

Reviewer 46Uu (score: 6) The main concerns focused on the scope of Assumption 1 and the presentation of the theoretical results. The rebuttal clarified the applicability of the invariance assumption across different domains, which directly addresses these points. I expect this reviewer would likely maintain their current score, or possibly increase it slightly, but not substantially change their overall assessment.

Reviewer zSdB (score: 2) This review primarily raised issues related to notation clarity, definition of variables, and presentation rather than the core technical contributions. While the rebuttal clarified several definitions and experimental details, the reviewer did not engage further in the discussion. It is therefore difficult to predict whether the score would change.

Reviewer ZYHS (score: 4) The main concerns were about the realism of the experimental setup and robustness to real-world settings. In the rebuttal, the authors added experiments on public real-world MRI and sparse CT datasets, which directly address these concerns. The reviewer responded that they would maintain their score.

Reviewer vQFi (score: 4) This reviewer raised concerns about the strength of the theoretical assumptions, computational overhead, and experimental realism. The rebuttal addressed scalability by clarifying computational costs relative to EI and added real-data experiments. While these responses mitigate some practical concerns, the theoretical assumptions remain unchanged. I expect this reviewer would likely maintain a weak-reject score or possibly move slightly upward.

---

### Decision · Program_Chairs · 2026-01-26

Accept (Poster)